# A human model of Batten disease shows role of CLN3 in phagocytosis at the photoreceptor–RPE interface

Cynthia Tang[1,8], Jimin Han[1,8], Sonal Dalvi[1], Kannan Manian[1], Lauren Winschel[1], Stefanie Volland[2], Celia A. Soto [1], Chad A. Galloway [1], Whitney Spencer[1], Michael Roll[1], Caroline Milliner[3], Vera L. Bonilha[3], Tyler B. Johnson[4], Lisa Latchney[1], Jill M. Weimer[4], Erika F. Augustine[5], Jonathan W. Mink[5], Vamsi K. Gullapalli[1], Mina Chung[1,6], David S. Williams [2] & Ruchira Singh [1,6,7✉]

Mutations in *CLN3* lead to photoreceptor cell loss in CLN3 disease, a lysosomal storage disorder characterized by childhood-onset vision loss, neurological impairment, and premature death. However, how *CLN3* mutations cause photoreceptor cell death is not known. Here, we show that CLN3 is required for phagocytosis of photoreceptor outer segment (POS) by retinal pigment epithelium (RPE) cells, a cellular process essential for photoreceptor survival. Specifically, a proportion of CLN3 in human, mouse, and iPSC-RPE cells localized to RPE microvilli, the site of POS phagocytosis. Furthermore, patient-derived CLN3 disease iPSC-RPE cells showed decreased RPE microvilli density and reduced POS binding and ingestion. Notably, POS phagocytosis defect in CLN3 disease iPSC-RPE cells could be rescued by wild-type *CLN3* gene supplementation. Altogether, these results illustrate a novel role of CLN3 in regulating POS phagocytosis and suggest a contribution of primary RPE dysfunction for photoreceptor cell loss in CLN3 disease that can be targeted by gene therapy.

[1] Department of Ophthalmology and Biomedical Genetics, University of Rochester, Rochester, NY, USA. [2] Department of Ophthalmology, Stein Eye Institute, Department of Neurobiology, David Geffen School of Medicine, Molecular Biology Institute, Brain Research Institute, University of California, Los Angeles, CA, USA. [3] Department of Ophthalmic Research, Cleveland Clinic, Cleveland, OH, USA. [4] Sanford Research, Sioux Falls, SD, USA. [5] Department of Neurology, University of Rochester, Rochester, NY, USA. [6] Center for Visual Science, University of Rochester, Rochester, NY, USA. [7] UR Stem Cell and Regenerative Medicine Center, University of Rochester, Rochester, NY, USA. [8]These authors contributed equally: Cynthia Tang, Jimin Han. ✉email: ruchira_singh@urmc.rochester.edu

Neuronal ceroid lipofuscinoses (NCLs) describes a group of genetically distinct neurodegenerative lysosomal storage diseases that involve excessive accumulation of lipofuscin. Juvenile NCL (JNCL) can be caused by mutations in *CLN3* (CLN3-Batten, CLN3 disease). CLN3 disease, the most common form of NCL, presents in early childhood with vision loss as the first clinical feature, followed some years later by progressive neurological dysfunction and ultimately premature death[1–7]. Although it is well established that retinal damage is responsible for vision loss in CLN3 disease, the primary cellular and molecular mechanisms leading to retinal degeneration in CLN3 disease are not known. This is partly due to limited and conflicting data on CLN3 localization and function in the retina[2,8–11], and lack of a suitable model system that recapitulates the human disease phenotype. Furthermore, clinical and histopathologic studies have shown the involvement of multiple retinal cell layers in CLN3 pathology, suggesting a potentially complex etiology[12–14]. Specifically, the accumulation of autofluorescent lipopigment (lipofuscin) in retinal neurons and degeneration of multiple retinal cell layers has been documented in CLN3 disease[13,14].

High levels of lipofuscin are a characteristic of the retinal pigment epithelium (RPE) in many macular dystrophies[15–17]. Increased lipofuscin has also been shown experimentally to accumulate in mouse models of defective photoreceptor outer segment (POS) phagosome degradation[18,19]. However, the RPE in CLN3 disease has notably low levels of lipofuscin[13,14], even though it still undergoes atrophy[13]. One explanation for this apparent paradox is that loss of vision in CLN3 disease patients starts at a young age (5–10 years old[5]), with reduced rod and cone responses[2,20,21] and photoreceptor cell loss[2,21,22], and that the low levels of lipofuscin in the RPE result from the presence of fewer photoreceptors. Each mammalian RPE cell ingests and degrades 10% of the distal POS disks on a daily basis[23], and lipofuscin accumulates normally with age as a result of POS degradation products[24]. Fewer photoreceptors mean fewer POS phagosomes, resulting in reduced lipofuscin accumulation[14,25,26]. Because of the early onset of disease, it is not clear whether the photoreceptor cell loss precedes any changes in the RPE, as would be predicted by this explanation. An alternative explanation for reduced RPE lipofuscin is that there is another defect associated with the RPE, besides lysosomal dysfunction. For example, it is plausible that RPE cells fail to take up POS in CLN3 disease. This hypothesis could also explain the increased autofluorescence accumulation observed in the photoreceptor layer (presence of POS debris) and photoreceptor loss in CLN3 disease[1,21,27,28]. Indeed, decreased uptake of POS by RPE cells in a form of retinitis pigmentosa caused by mutations in the *MERTK* gene[29,30] leads to a similar pathology as CLN3 disease. However, apart from one published study in a mouse model[31], the role of RPE cell dysfunction in CLN3 disease retinal pathology has not been investigated.

The human induced pluripotent stem cell (hiPSC) technology allows the investigation of pathological and molecular changes in an individual cell type, using cells derived from patients. With regard to human retinal diseases, the use of hiPSCs is specifically pertinent to RPE-based disorders[32–35]. Despite the fact that hiPSC-RPE monocultures lack the complexity of functional and structural interactions with photoreceptors, they have been successfully used to investigate the pathological mechanisms of both early onset retinal diseases, such as Best disease[36] and late onset disease, such as age-related macular degeneration[37,38]. In fact, several studies have now shown that feeding a physiological amount of POS to hiPSC-RPE cells can be utilized to investigate POS phagocytosis regulation in normal versus diseased cells[36,39,40].

In this study, using primary mouse and human RPE, CLN3 overexpression, and hiPSC-based disease modeling experiments, we show that a proportion of CLN3 in RPE cells is localized to the RPE apical microvilli. Notably, cell-autonomous CLN3 dysfunction in RPE cells is sufficient to affect RPE microvillar density and POS binding and consequently POS uptake by the CLN3 disease hiPSC-RPE cells, leading to decreased accumulation of autofluorescent POS-digestion products in the CLN3 disease hiPSC-RPE cells. This result, together with longitudinal multimodal imaging of the retina in a CLN3 disease patient, suggests that autofluorescent changes in the photoreceptor–RPE complex that are concordant with POS phagocytosis defect may precede photoreceptor cell loss in CLN3 disease. Importantly, lentivirus-mediated overexpression of wild-type CLN3 can rescue POS phagocytosis defect in the CLN3 disease hiPSC-RPE cells. Altogether, our data define a previously unidentified role of CLN3 at the photoreceptor–RPE interface and suggest a contribution of primary RPE dysfunction in instigating reduced RPE lipofuscin that can be therapeutically targeted by gene augmentation.

## Results

**Chronic POS feeding leads to reduced autofluorescent material accumulation in CLN3 disease hiPSC-RPE compared to parallel cultures of control hiPSC-RPE.** Histopathological studies on human cadaver eyes from donors with CLN3 disease have shown reduced lipofuscin compared to unaffected control individuals[1,13,14]. However, the mechanism underlying reduced lipofuscin accumulation in CLN3 disease RPE cells is not known. In an attempt to recapitulate this characteristic RPE phenotype and isolate the plausible contribution of RPE dysfunction in CLN3 disease, hiPSCs from control (unaffected heterozygote family member, unrelated healthy subject) and 2 distinct CLN3 patients harboring the common 966 bp homozygous deletion in *CLN3*[41] were differentiated to RPE cells in culture and analyzed. Control and CLN3 disease hiPSC-RPE display several typical RPE characteristics including expected localization of RPE-signature proteins: Zonula Occludens-1 (ZO-1: tight junctions, Fig. 1A), Ezrin (EZR; apical, Fig. 1B) and Bestrophin-1 (BEST1: basolateral, Fig. 1C), expression of several RPE-signature genes/proteins: BEST1, Cellular Retinaldehyde-Binding Protein (CRALBP), EZR, MER proto-oncogene, Tyrosine Kinase (MERTK), Melanocyte Inducing Transcription Factor (MITF), Occludin (OCLN), Pigment Epithelium-Derived Factor (PEDF), and Retinoid Isomerohydrolase (RPE65) (Fig. 1D, E); and formation of functional tight junctions with transepithelial resistance (TER) comparable to human RPE cells in vivo (Fig. 1F)[42,43]. Furthermore, consistent with a polarized RPE monolayer, control and CLN3 hiPSC-RPE formed fluid domes when cultured on non-permeable plastic support (Supplementary Fig. 1). Overall, control and CLN3 hiPSC-RPE showed several baseline characteristics akin to in vivo human RPE monolayer.

Given that autofluorescent material accumulation in RPE cells is a consequence of POS uptake and digestion, we next evaluated if daily feeding of a physiological dose of POS for a prolonged duration would be sufficient to mimic the patterns of RPE autofluorescence accumulation seen in control versus CLN3 disease donor eyes (Fig. 2A)[1,13,14]. Of note, autofluorescence accumulation was analyzed in the spectral range that is consistent with lipofuscin accumulation in human retina/RPE cells[24,44]. Specifically, confocal microscopy analyses of retina sections and RPE wholemounts showed increased autofluorescence material/debris accumulation in the photoreceptor layer and reduced autofluorescence in RPE cells from CLN3 disease donor eyes compared to RPE autofluorescence in non-CLN3 donor eyes, an age-matched donor eye with Charcot Marie Tooth Disease and

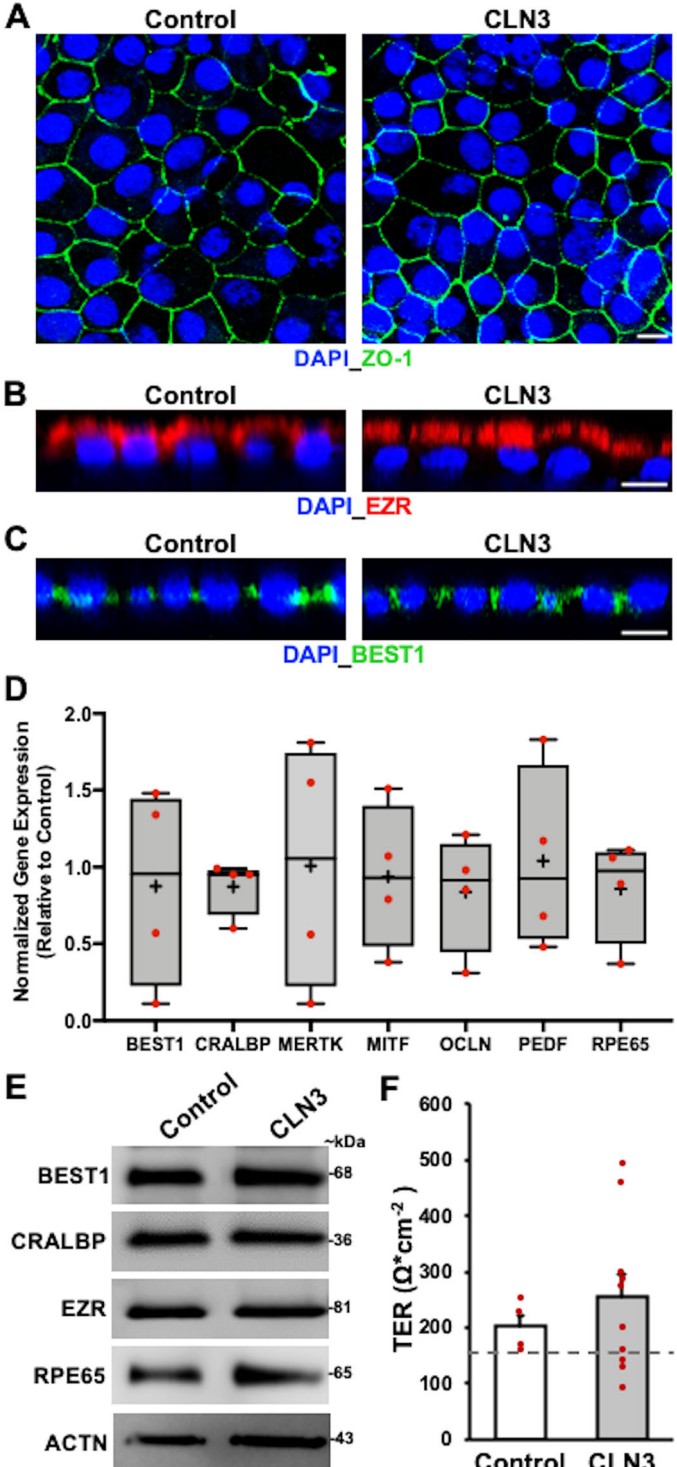

adult healthy control donor eye (Fig. 2A, B; Supplementary Fig. 2). Importantly, the reduced RPE autofluorescence in the CLN3 disease donor eye was not a consequence of RPE atrophy/ cell death, as decreased RPE autofluorescence was also observed in areas with intact RPE monolayer (Fig. 2B). Next, a previously established protocol[36,45] was utilized to evaluate the abundance of POS-digestion products or autofluorescence accumulation in control versus CLN3 disease hiPSC-RPE cells (Fig. 2C). Analysis of autofluorescence material post-chronic POS feeding (Fig. 2C), showed that consistent with pathology of human CLN3 disease

RPE, the amount of autofluorescence was decreased in CLN3 disease hiPSC-RPE cells compared to parallel cultures of control hiPSC-RPE cells (Fig. 2D–H). Note that control and CLN3 disease hiPSC-RPE display similar localization of tight junction marker, ZO-1 (Fig. 2D). In contrast, the amount of autofluor-escence was decreased in CLN3 disease hiPSC-RPE cells (Fig. 2D–H). Furthermore, orthogonal view of hiPSC-RPE sections confirmed that autofluorescence accumulation post-chronic POS feeding was within the control and CLN3 disease hiPSC-RPE cells (Fig. 2F). Of note, parallel cultures of control

**Fig. 1 Baseline RPE characteristics are similar in control and CLN3 disease hiPSC-RPE cells. A–C** Representative confocal microscopy images showing similar and expected localization of tight junction marker, Zonula Occludens-1 (ZO-1) (**A**), apically localized RPE protein, Ezrin (EZR) (**B**), and basolaterally expressed RPE protein, Bestrophin-1 (BEST1) (**C**), in control and CLN3 disease hiPSC-RPE cells. Of note, cell nuclei are stained with DAPI (Scale bar = 10 µm) (n ≥ 3). **D** Quantitative real-time PCR analyses showing similar expression of RPE-signature genes in CLN3 disease hiPSC-RPE cells relative to control hiPSC-RPE cells: BEST1 ($p = 0.76$), Cellular Retinaldehyde-Binding Protein (CRALBP; $p = 0.30$), MER proto-oncogene (MERTK; $p = 0.98$), Microphthalmia-associated Transcription Factor (MITF; $p = 0.83$), Occludin (OCLN; $p = 0.50$), Pigment Epithelium-Derived Factor (PEDF; $p = 0.91$), Retinoid Isomerohydrolase RPE65 ($p = 0.51$). GAPDH served as loading control. (Control: $n = 3$, CLN3: $n = 4$). Of note, in the boxplots, + represents mean, center line represents median, box represents interquartile range between first and third quartiles, and whiskers represent 1.5* interquartile range.
**E** Representative Western blot images showing similar expression of RPE-signature proteins ($n = 3$): BEST1 (68 kDa), CRALBP (36 kDa), EZR (81 kDa), RPE65 (65 kDa), and Actin (ACTN, 43 kDa). **F** Transepithelial resistance (TER) measurements showing the presence of functional tight junctions with TER similar to the reported in vivo threshold of ~150 $\Omega$ cm$^{-2}$[42,43] (dotted line) in both control and CLN3 disease hiPSC-RPE cells grown as a monolayer on Transwell inserts (Control: $n = 4$, CLN3: $n = 10$; $p = 0.27$). For all graphs in Fig. 1, statistical significance was determined using two-tailed unpaired Student's t-test.

and CLN3 disease hiPSC-RPE at a similar age (~60–90 days in culture) displayed no autofluorescence in the absence of POS feeding (Supplementary Fig. 3).

Altogether, these results show that decreased autofluorescence accumulation, a known pathological characteristic of CLN3 disease RPE in vivo, can be mimicked in patient-derived hiPSC-RPE cells. Furthermore, the ability to recapitulate the difference in autofluorescence material accumulation between control and CLN3 disease hiPSC-RPE cells after prolonged exposure to unaffected (no *CLN3* mutation) POS, suggests that RPE autonomous CLN3 dysfunction is sufficient to instigate the reduced RPE autofluorescence observed in CLN3 disease eyes.

**Decreased POS phagocytosis by CLN3 disease hiPSC-RPE compared to control hiPSC-RPE cells.** Decreased autofluorescent material accumulation in CLN3 disease hiPSC-RPE after chronic POS-feeding (Fig. 2) implicated impaired POS uptake by CLN3 disease hiPSC-RPE cells. Therefore, to confirm a POS phagocytosis defect, we next performed acute experiments to compare POS uptake[34,36,45] by control versus CLN3 disease hiPSC-RPE cells. Briefly, FITC-labeled or unlabeled POS (~20–40 POS/RPE cell) were fed to parallel cultures of control and CLN3 disease hiPSC-RPE for 2 h and POS uptake was quantified by measuring the levels of FITC or rhodopsin (RHO) (a POS-specific protein) within the hiPSC-RPE cells (Fig. 3A) (see Methods section for further details). Confocal microscopy analyses of FITC fluorescence in control and CLN3 disease hiPSC-RPE cells after POS feeding showed a reduced number of total FITC-POS (bound+ingested) in CLN3 disease hiPSC-RPE cells compared to control hiPSC-RPE cells (Fig. 3B–E). Furthermore, Western blot analyses of unlabeled POS uptake (bound+ingested) through quantification of RHO, a POS-specific protein (Fig. 4A), showed reduced RHO levels post-POS feeding in CLN3 disease hiPSC-RPE cells compared to control hiPSC-RPE cells (Fig. 4B–G). Notably, reduced RHO/POS uptake by CLN3 disease hiPSC-RPE cells was seen when comparing parallel cultures of both (i) young control and CLN3 hiPSC-RPE cells that were cultured for ~20–40 days (~D20-40) (Fig. 4B–D) and (ii) aged control and CLN3 hiPSC-RPE cells (~D60-D150; Fig. 4E–G). Overall, the decreased amount of both FITC-POS and RHO in CLN3 hiPSC-RPE cells compared to control hiPSC-RPE cells post-POS feeding suggests that CLN3 is required for efficient POS phagocytosis by hiPSC-RPE cells.

**CLN3 regulates POS binding by hiPSC-RPE cells.** Having established a POS phagocytosis defect in CLN3 disease hiPSC-RPE cells, we next evaluated whether disease-causing mutations in *CLN3* preferentially affect POS binding versus POS internalization by hiPSC-RPE cells. Orthogonal views of FITC-POS

fed control and CLN3 disease hiPSC-RPE wholemounts, immunostained for cell nuclei (DAPI) and tight junction protein (ZO-1), showed a reduced number of FITC-POS both apically and basally relative to ZO-1 localization (Supplementary Fig. 4). Using a previously published protocol[46], analyses of FITC-POS localized apically (bound) versus basally (internalized) relative to ZO-1, showed reduced numbers of both bound and internalized POS in CLN3 disease hiPSC-RPE cells compared to control hiPSC-RPE cells (Supplementary Fig. 4). In contrast, the number of internalized FITC-POS normalized to number of bound FITC-POS was not different in control versus CLN3 disease hiPSC-RPE cells (Supplementary Fig. 4). Overall, these results suggest a POS-binding defect in CLN3 disease hiPSC-RPE cells. To further confirm a POS-binding defect in CLN3 disease hiPSC-RPE cells, we next evaluated the amount of bound-POS in control versus CLN3 disease hiPSC-RPE cells 30 min post-POS feeding (Fig. 5A–E). Specifically, control and CLN3 disease hiPSC-RPE cells were fed ~20–40 FITC-POS/RPE cell for 30 min at 17 °C (Fig. 5A), a temperature conducive to POS binding but not POS internalization[47]. Next, the number of FITC-POS (bound) was compared between control and CLN3 disease hiPSC-RPE cells (Fig. 5B–E). Quantification of FITC-POS after immunocytochemical analyses revealed a lower number of bound-FITC-POS in CLN3 disease hiPSC-RPE cells compared to control hiPSC-RPE cells (Fig. 5B–E). Of note, consistent with preferential POS-binding and limited POS uptake at 17 °C, orthogonal view of hiPSC-RPE fed FITC-POS for 30 min at 17 °C, and immunostained for cell nuclei (DAPI), and tight junction protein (ZO-1), showed FITC-POS expression predominantly apically to ZO-1 staining in both control and CLN3 disease hiPSC-RPE cells (Supplementary Fig. 5). Interestingly, quantitative Western blot analyses showed no difference in the levels of POS engulfment receptor (MERTK) and POS binding receptor (Integrin Subunit Alpha V (ITGAV), Integrin Subunit Beta 5 (ITGB5)) between control and CLN3 disease hiPSC-RPE cells (Supplementary Fig. 6). With a recent study showing abnormal F-actin cytoskeleton reduces the POS binding in stem cell-derived cultures[46] and the previously reported finding that CLN3 deficiency has been shown to cause abnormally organized actin and cytoskeleton architecture in other cell type(s)[48], we next evaluated F-actin morphology in control versus CLN3 disease hiPSC-RPE cultures (Fig. 6A, B). Localization of F-actin using phalloidin staining showed the expected and similar localization of F-actin in control versus CLN3 disease hiPSC-RPE cultures (Fig. 6A). Furthermore, orthogonal view of RPE cultures revealed apically localized F-actin in both control and CLN3 disease hiPSC-RPE cultures (Fig. 6B). Although no overt defects in F-actin localization were seen in CLN3 disease hiPSC-RPE cells, transmission electron microscopy (TEM) analyses of control and CLN3 disease hiPSC-RPE cells showed reduced density of apical microvilli in CLN3

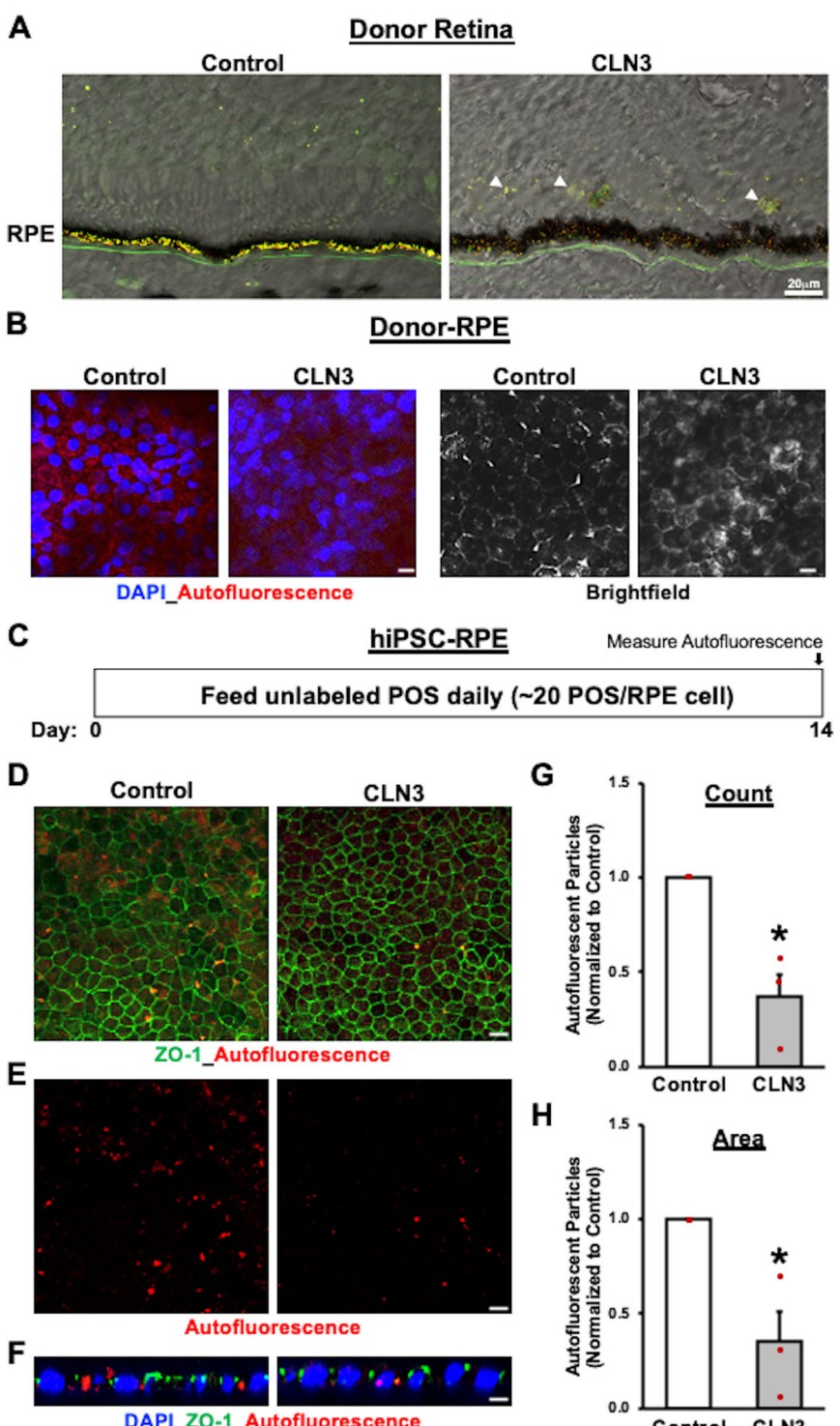

disease hiPSC-RPE cells compared to control hiPSC-RPE cells (Fig. 6C, D). In contrast, no difference in microvilli width (Fig. 6C, E) and length (Fig. 6C, F) was seen between control and CLN3 disease hiPSC-RPE cells.

Altogether, these results suggest that CLN3 is required for RPE apical microvilli homeostasis and POS binding by human RPE cells. Furthermore, because RPE apical microvilli density affects POS binding by RPE cells[46], it is plausible that the loss of apical microvilli contributes to reduced POS binding by CLN3 disease hiPSC-RPE cells.

**A proportion of CLN3 is localized in the RPE microvilli in both native human and mouse tissue and hiPSC-RPE cells.** Given the data demonstrating a role of CLN3 in RPE apical microvilli homeostasis and POS binding by hiPSC-RPE cells (Figs. 5, 6), we investigated the localization of CLN3 within RPE cells. Of note, although CLN3 has never been reported to reside in RPE microvilli, CLN3 was shown to be present in the microvilli of the Malpighian tubules in a *Drosophila* model[49]. Immunocyto-chemical analyses showed apical localization of CLN3 in control hiPSC-RPE cells and co-localization of CLN3 with an RPE

**Fig. 2 Decreased accumulation of autofluorescent material after chronic POS-feeding in CLN3 disease hiPSC-RPE cells mimics reduced presence of autofluorescent POS-digestion products in CLN3 disease donor RPE monolayer. A** Autofluorescence overlying brightfield images in cryosections of mid-periphery fragments of retina-RPE-choroids obtained from a CLN3 disease donor and age-matched control eye showed decreased accumulation of autofluorescent material in the RPE layer of CLN3 disease donor eyes in spectral wavelength consistent with lipofuscin (red and green channels). Additionally, increased accumulation of autofluorescent debris is seen in the photoreceptor layer (indicated by white arrowheads) in the CLN3 disease donor retina compared to control retina. (Scale bar = 20 μm) (n = 1). **B** Confocal microscopy analyses of RPE wholemounts obtained from the same CLN3 disease and control donor eyes (as were shown in panel **A**) showed decreased autofluorescence (measured in red channel) in areas with intact RPE monolayer. Of note, light microscopy images (right panel) are from the same area of view as autofluorescence images (left panel) (scale bar = 10 μm) (n = 1). Furthermore, to use age-matched control, control cadaver RPE used in panels **A**, **B** was from a donor with Charcot Marie Tooth Disease. **C** Schematic showing the experimental design used to assess autofluorescence material accumulation in hiPSC-RPE cells with chronic POS-feeding. As shown, hiPSC-RPE were fed a physiological dose of unlabeled POS (~20 POS/RPE cell) daily for 14 days. Subsequently, the cells were fixed, immunostained and imaged for autofluorescent material accumulation (red channel) using confocal microscopy. **D–F** Consistent with CLN3 disease donor RPE data (**A**, **B**), confocal microscopy analyses after 2 weeks of daily POS feeding showed decreased autofluorescent material (ex: 546 nm, em: 560–615 nm) in CLN3 disease hiPSC-RPE monolayer compared to control hiPSC-RPE monolayer. In contrast, similar localization of tight junction protein (ZO-1) was seen in both control and CLN3 hiPSC-RPE monolayers. Of note, panel (**E**) shows the same image region of control and CLN3 disease hiPSC-RPE as panel **D** but displays autofluorescence material accumulation without ZO-1 staining (scale bar = 10 μm) (n = 3). **F** Representative orthogonal view of confocal z-stack images showing that autofluorescent material localizes basal to the tight junction marker, ZO-1, and is within the control and CLN3 disease hiPSC-RPE cells. Cell nuclei is stained with DAPI (scale bar = 10 μm) (n = 3). **G, H** Quantitative analyses showing both decreased count (**G**, p = 0.011) and area (**H**, p = 0.026) of accumulated autofluorescent material (ex: 546 nm, em: 560–615 nm) after 14 days of consecutive POS feeding in parallel cultures of control versus CLN3 disease hiPSC-RPE cells (n = 3). Statistical significance was determined using two-tailed unpaired Student's t-test. *p < 0.05.

microvilli marker, EZR (Fig. 7A). To further confirm CLN3 localization to hiPSC-RPE microvilli, we next analyzed control hiPSC-RPE cells using subcellular fractionation. We isolated RPE microvilli from control hiPSC-RPE cells using a published protocol that utilized lectin-agarose beads[50]. Consistent with our immunocytochemical data showing EZR and CLN3 co-localization (Fig. 7A), confocal microscopy analyses of the microvilli fraction adhering to the lectin-agarose beads showed the presence and co-localization of EZR and CLN3 (Fig. 7B). Furthermore, consistent with previous proteomics data of RPE microvilli[50] no expression of laminin (LAM) was seen on microvilli-containing lectin-agarose beads (Supplementary Fig. 7).

To further validate CLN3 localization in RPE microvilli, we also utilized in vitro lentivirus-mediated overexpression of human *CLN3* using pHIV-MYC-CLN3-IRES-EGFP and pHIV-FLAG-CLN3-IRES-EGFP vectors (Supplementary Fig. 8). Consistent with robust *CLN3* overexpression, transduction of 293FT and control hiPSC-RPE cells with lentiviral vectors (*i*) pHIV-MYC-CLN3-IRES-EGFP and (*ii*) pHIV-FLAG-CLN3-IRES-EGFP resulted in prominent EGFP expression in both 293FT and hiPSC-RPE cells (Fig. 7C, D; Supplementary Fig. 8). Of note, diffusion across and accumulation of EGFP in the nucleus after lentiviral transduction (Fig. 7C, Supplementary Fig. 8) is because the bi-cistronic vector pHIV-MYC-CLN3-IRES-EGFP and pHIV-FLAG-CLN3-IRES-EGFP containing CLN3 and EGFP are translated separately within the single mRNA at the internal ribosome entry site (IRES)[51,52]. Furthermore, the presence of EGFP in the nucleus is due to its low molecular weight (~27 kDa) and the nuclear and kinetic entrapment of EGFP homomultimers[53]. Western blot analyses of untransduced versus transduced 293FT cells showed selective expression of MYC and FLAG in transduced cells (Supplementary Fig. 8). Also, consistent with anti-CLN3 antibody data, immunocytochemical analyses of MYC-CLN3 (Supplementary Fig. 9) and FLAG-CLN3 (Fig. 7D) in transduced cells using (*i*) anti-MYC and anti-EZR antibody (Supplementary Fig. 9) and (*ii*) anti-FLAG and anti-EZR antibody (Fig. 7D) showed co-localization of EZR and MYC-CLN3 (Supplementary Fig. 9) and EZR and FLAG-CLN3 (Fig. 7D). Also consistent with prior studies[54,55], co-localization of FLAG-CLN3 and lysosomal marker, LAMP1, was seen in control hiPSC-RPE cells transduced with pHIV-FLAG-CLN3-IRES-EGFP vector (Supplementary Fig. 9).

To validate the CLN3 localization data in hiPSC-RPE cells (Fig. 7A–D, Supplementary Fig. 9), we next examined primary human RPE and mouse RPE cells. Similar to hiPSC-RPE, wholemounts and sections of primary adult human RPE and mouse RPE displayed co-localization of EZR and CLN3 (Fig. 7E, F and Supplementary Fig. 10). Of note, for immunocytochemical analyses, due to the debated specificity of previously used CLN3 antibodies[11], we utilized (*i*) CLN3 blocking peptide (Supplementary Fig. 10), (*ii*) tissue from CLN3 knockout mice[56] (Fig. 7E) and (*iii*) no primary antibody control (Fig. 7E). Of note, because we used a mouse-CLN3 antibody in our experiments, we also utilized a commercial kit (mouse on mouse or MOM kit, Vector Lab Inc. FMK-2201) to overcome the non-specific background due to endogenous IgG staining[57]. Furthermore, the utility of retinal/RPE tissue from the CLN3 knockout mice[56] in the current study was limited to antibody verification for immunolocalization experiments.

Western blot analyses of endogenous CLN3 in RPE cells showed a distinct pattern of CLN3 in mouse versus human RPE cells (Fig. 7G). Specifically, when protein lysate from native mouse RPE (~15–20 μg total protein) was run on SDS-PAGE, we observed multiple bands (possibly due to non-specific background due to use of the mouse CLN3 antibody). However, the expected broad band between ~50–60 kDa[58] was also seen (Fig. 7G). In contrast, when protein lysates from human RPE (primary, hiPSC) were analyzed by Western blotting with the same CLN3 antibody, we observed a strong band ~50 kDa (when 15–20 μg total protein was loaded on gels) (Fig. 7G). However, in some experiments when a higher amount of total protein (30 μg) was loaded in gels, we did see an additional band consistent with CLN3 glycosylation (Supplementary Fig. 10). Of note, mouse and human CLN3 have previously been shown to run differently on Western blots[58,59] and differential glycosylation has been seen based on species as well as the tissue source[58,60]. Furthermore, incubation of CLN3 primary antibody with CLN3 blocking peptide in Western blotting experiments inhibited CLN3 reactivity in primary human RPE and hiPSC-RPE samples (Supplementary Fig. 10). However, to further confirm that the mouse CLN3 antibody used in our experiments recognizes CLN3, we evaluated recognition of overexpressed CLN3 protein by the CLN3 antibody via Western blotting (Supplementary Fig. 10). Specifically, Western blot analyses of hiPSC-RPE cells transduced with pHIV-MYC-CLN3-IRES-EGFP lentiviral vector for ~5 days

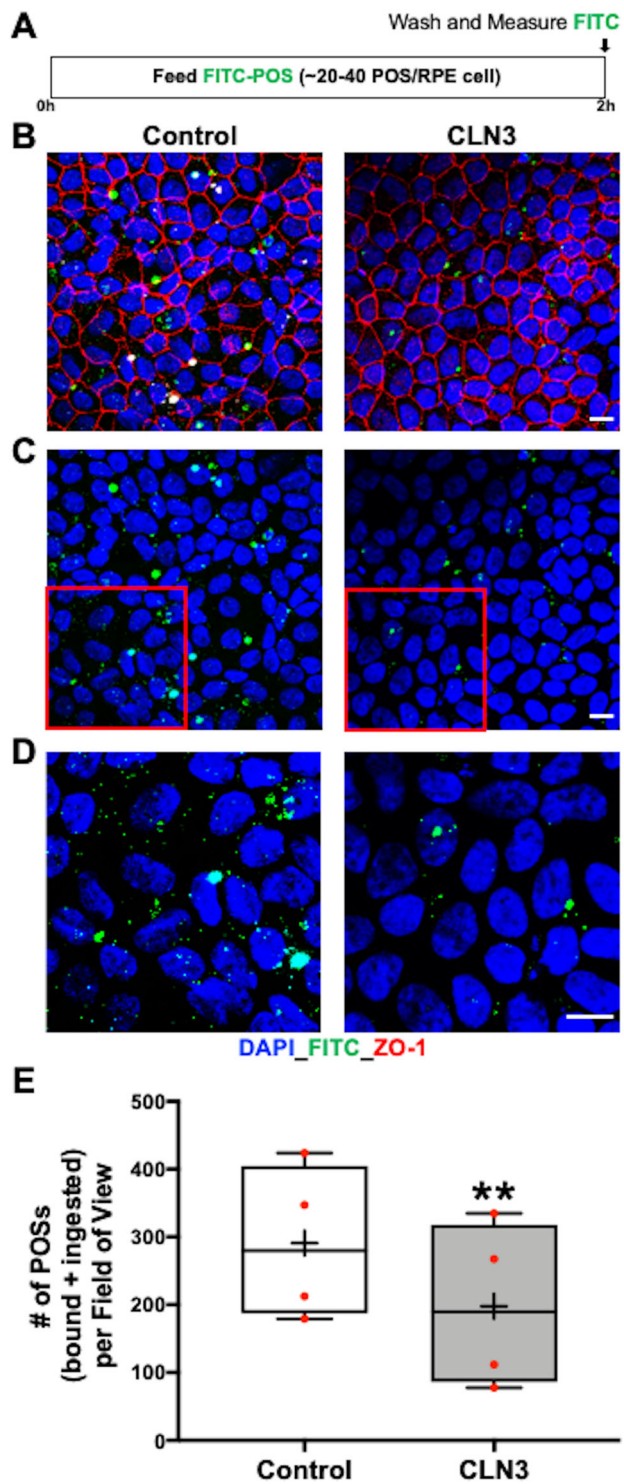

**Fig. 3 Decreased phagocytosis of FITC-POS by CLN3 disease hiPSC-RPE cells. A** Schematic showing the protocol used to evaluate POS phagocytosis by hiPSC-RPE cells. Specifically, hiPSC-RPE cells were fed ~20–40 FITC-labeled POS/RPE cell for 2 h at 37 °C. Subsequently, any FITC-POS remaining on hiPSC-RPE cell surface was removed by washing with 1X PBS and the cells were fixed, immunostained with ZO-1 (red channel) and DAPI (blue channel) and the amount of POS phagocytosed by hiPSC-RPE cells was determined by measuring FITC fluorescence (green channel) using confocal microscopy. **B–D** Representative confocal microscopy images showing a similar pattern of ZO-1 (**B**) and DAPI localization (**B–D**), but decreased number of FITC-POS in CLN3 disease hiPSC-RPE cultures compared to control hiPSC-RPE cultures 2 h post-FITC-POS feeding (scale bar = 10 μm) ($n = 4$). Of note, panel **D** is the enlarged view of the highlighted area (red box) in panel **C**. **E** Quantitative analyses of FITC-fluorescence-labeled POS particles (particles < 5 μm, threshold set to exclude POS aggregates), 2 h post-POS-feeding, showing decreased number of phagocytosed (bound+ingested) FITC-POS per field of view in parallel cultures of CLN3 disease hiPSC-RPE cells compared to control hiPSC-RPE cells ($n = 4$, $p = 0.003$, two-tailed unpaired Student's $t$-test). For the boxplot, + represents mean, center line represents median, box represents interquartile range between first and third quartiles, and whiskers represent 1.5* interquartile range. **$p < 0.005$.

showed ~6.43-fold higher amount of CLN3 in the transduced hiPSC-RPE cells (Supplementary Fig. 10). Of note, we did not observe a difference in the size of endogenous versus over-expressed CLN3 protein by Western blot analyses as in hiPSC-RPE cells transduced cells with bi-cistronic pHIV-MYC-CLN3-IRES-EGFP vector, the CLN3 and EGFP are translated separately within the single mRNA at the IRES.

Consistent with our immunocytochemical data (Fig. 7A, B), Western blot analyses of the RPE microvilli versus cell pellet fraction showed the presence of (*i*) CLN3 in both the RPE microvilli and the cell pellet fraction (Supplementary

Fig. 10) and (*ii*) majority of EZR, an RPE microvilli protein, in the microvilli fraction (Supplementary Fig. 10). Altogether, using immunocytochemical analyses, subcellular fractionation assays, and gene-overexpression studies, our data suggest that a proportion of CLN3 in RPE cells localizes to the RPE apical microvilli.

**Overexpression of wild-type CLN3 in CLN3 disease hiPSC-RPE rescues the POS phagocytosis defect.** Previous studies on the impact of CLN3 disease-causing mutations, including the homozygous 966 bp deletion spanning exon 7 and exon 8 that we used in the current study are contradictory, suggesting both loss of protein function[61] and functionally viable CLN3[62]. Consistent with previously published studies[61,63], quantitative real-time PCR analyses showed reduced expression of the endogenous *CLN3* gene transcript in control versus CLN3 disease hiPSC-RPE cultures (Fig. 8A). Remarkably, lentivirus-mediated overexpression of wild-type (WT)-CLN3 (pHIV-MYC-CLN3-IRES-EGFP or pHIV-FLAG-CLN3-IRES-EGFP) in CLN3 disease hiPSC-RPE cells for ~5–6 days (Fig. 8B–D) had no negative effect on cell viability. Similar to all other POS phagocytosis experiments in this study (Figs. 3–5, Supplementary Fig. 4, 5), untransduced CLN3 disease and WT-CLN3 transduced CLN3 disease hiPSC-RPE cultures displayed TER > 150 $\Omega \cdot cm^{-2}$ (Fig. 8C) with the expected and proper localization of tight junction protein, ZO-1 (Fig. 8D). Importantly, WT-CLN3 expression via lentiviral-mediated transduction was sufficient to increase the amount of POS phagocytosis by CLN3 disease hiPSC-RPE cultures (Fig. 8E–G). Specifically, with regard to POS phagocytosis, hiPSC-RPE cells from both CLN3 disease patient hiPSC lines used in the current study with the common 966 bp deletion spanning exon 7 and 8 when trans-duced with either pHIV-MYC-CLN3-IRES-EGFP or pHIV-FLAG-CLN3-IRES-EGFP for 5 days displayed increased POS uptake, as measured by RHO levels post 2 h POS feeding, compared to parallel cultures of untransduced CLN3 disease hiPSC RPE cells (Fig. 8E–G). Furthermore, probing Western blots with anti-GFP antibody (Fig. 8F) showed the expected band (~27 kDa) in transduced CLN3 disease hiPSC-RPE cells. Of note, lentiviral-mediated transduction of WT-MYC-CLN3 or WT-FLAG-CLN3 had no impact on ACTN levels in CLN3

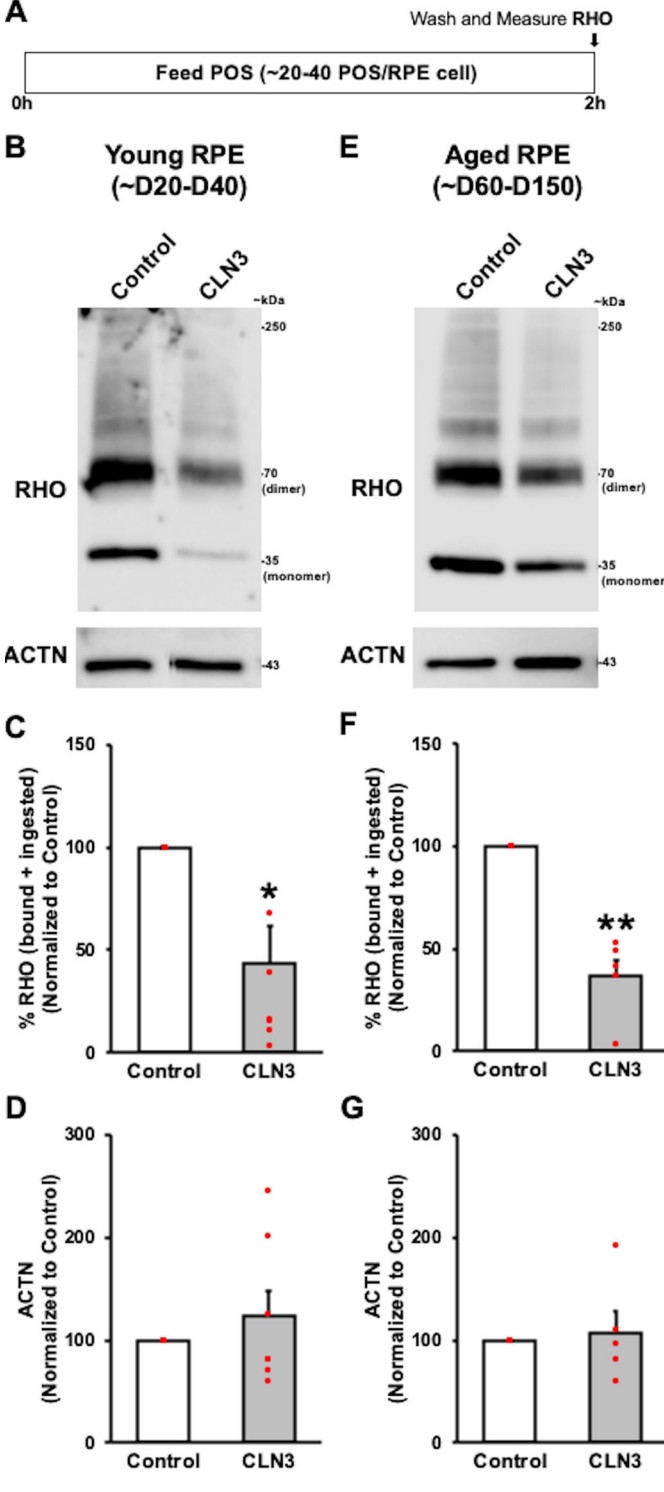

**Fig. 4 Reduced phagocytosis of unlabeled POS by CLN3 disease hiPSC-RPE cells. A** Schematic showing the protocol used to measure phagocytosis of unlabeled POS by hiPSC-RPE cells. Specifically, hiPSC-RPE cultures were fed ~20–40 POS/RPE cell for 2 h at 37 °C. Subsequently, any remaining POS on the surface of RPE cells was removed by washing with 1X PBS and the amount of phagocytosed (bound+ingested) POS was quantified by measuring the amount of Rhodopsin (RHO), a POS-specific protein, within the hiPSC-RPE cells. Of note, the total protein in the hiPSC-RPE cell lysate served as the normalization control in these experiments. **B–G** Representative Western blot images (**B**, **E**) and quantitative Western blot analyses (**C**, **F**) post 2 h POS-feeding showing reduced amount of RHO (monomer band 35 kDa, dimer band 70 kDa, and aggregate/multimer bands >70 kDa that are normally seen in RHO Western analyses[36,45]) relative to total protein in parallel cultures of both young (D20-50 in culture) (**B**, **C**; $p = 0.014$) and old (D60-150) (**E**, **F**; $p = 0.0019$) CLN3 disease hiPSC-RPE cells compared to control hiPSC-RPE cells. Of note, unlike RHO, no differences in Actin (ACTN) levels relative to total protein were seen between parallel cultures of both young (**B**, **D**; $p = 0.41$) and old (**E**, **G**; $p = 0.75$) control versus CLN3 disease hiPSC-RPE cells in these experiments. (Control: $n = 5$, CLN3: $n = 7$). For all graphs in Fig. 4, statistical significance was determined using two-tailed unpaired Student's $t$-test. *$p < 0.05$; **$p < 0.005$.

autofluorescent material in the photoreceptor layer and conversely lack of autofluorescent POS digestion products, lipofuscin, in RPE cells (Fig. 2A)[2]. The currently accepted theory for decreased lipofuscin in CLN3 disease cadaver RPE cells is that photoreceptor loss and therefore POS loss no longer obligates RPE cells to uptake POS[2,13]. Notably, in this scenario, autofluorescence changes in the photoreceptor–RPE layer would occur after photoreceptor cell loss. However, our data showing reduced POS binding and thereby decreased POS uptake (Figs. 3–5) and consequently decreased accumulation of autofluorescent POS-digestion products (Fig. 2) provides an alternate explanation for the lack of lipofuscin in the RPE layer in the CLN3 disease post-mortem; namely, reduced uptake of POS by RPE cells in the CLN3 disease retina. The decreased uptake of POS by CLN3 disease RPE cells can also explain the build-up of excess autofluorescence in the photoreceptor layer and subsequent photoreceptor cell death due to accumulation of POS debris in the CLN3 disease retina. This theory is consistent with electroretinography (ERG) recordings in CLN3 disease patients at the early stage of the disease that clearly document reduced rod-cone response prior to inner retina dysfunction[2,20,21].

A clinical correlate of these findings is shown by longitudinal multimodal imaging of a young CLN3 disease patient that is consistent with autofluorescent changes preceding POS loss (Fig. 9). A hallmark of CLN3 disease is bull's eye maculopathy (BEM), which describes a central area of atrophy, surrounded by concentric circles of diseased retina and then healthier retina peripherally. The margin of the BEM lesion, or junctional zone, represents the leading edge of disease progression along the retina. In contrast to retinal imaging analyses of a normal human retina (Fig. 9A–C), in a CLN3 disease patient eye at age 7, fundus photograph shows a BEM (Fig. 9D). In addition, at this timepoint, fundus autofluorescence (FAF) imaging displayed a hyperautofluorescent annulus at the margin of BEM (Fig. 9E) and optical coherence tomography (OCT) revealed central loss of the POS layer with POS preservation peripherally, corresponding to the location of the hyperautofluorescent annulus (Fig. 9F). Subsequently, at age 8 years, fundus photograph showed that the BEM has enlarged (Fig. 9G). Furthermore, FAF and OCT analysis at this time point showed that hyperautofluorescent annulus had also

disease hiPSC-RPE cells (Fig. 8E, G). Overall, these experiments suggest CLN3 loss of function in CLN3 disease hiPSC-RPE cells and demonstrate a rescue of POS phagocytosis defect by lentiviral-mediated expression of WT-CLN3 in CLN3 disease hiPSC-RPE cells.

**Structural and functional alterations in CLN3 disease retina in vivo are concordant with impaired POS phagocytosis and primary outer retina dysfunction**. Post-mortem histopathologic specimens from a CLN3 disease donor show accumulation of

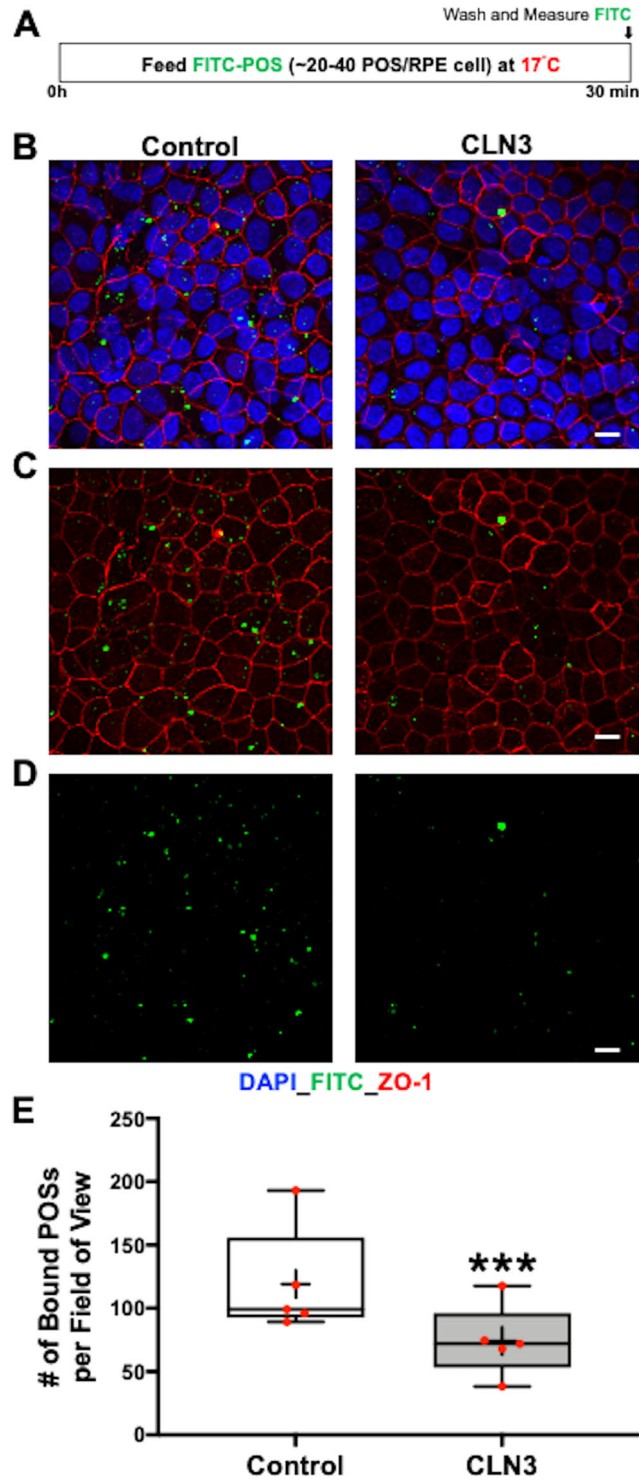

**Fig. 5 CLN3 disease hiPSC-RPE display a POS-binding defect.**
**A** Schematic showing the protocol used to evaluate POS binding by hiPSC cells. Specifically, hiPSC-RPE cells were fed FITC-POS (~20–40 POS/RPE cell) at 17 °C for 30 min, a temperature favorable to POS binding but not POS internalization[47]. Subsequently, POS-fed hiPSC-RPE cells were washed with 1X PBS to remove any POS remaining on the hiPSC-RPE cell surface. Next, the hiPSC-RPE cells were fixed, immunostained with ZO-1 (red channel) and DAPI (blue channel) and the amount of POS phagocytosed by hiPSC-RPE cells was determined by measuring FITC-fluorescence (green channel) using confocal microscopy. **B–D** Representative confocal microscopy images showing DAPI (**B**) ZO-1 (**B**, **C**), and FITC fluorescence (**B–D**) in parallel cultures of control versus CLN3 disease hiPSC-RPE cultures. Notably, reduced amount of FITC-fluorescence (bound-POS) post 30 min FITC-POS feeding at 17 °C is seen in CLN3 disease hiPSC-RPE cells compared to control hiPSC-RPE cells (scale bar = 10 μm) (n = 5).
**E** Quantitative analyses showed reduced number of bound FITC-POS particles (particles < 5 μm, threshold set to exclude POS aggregates) in CLN3 disease hiPSC-RPE cells compared to control cells (n = 5, p = 0.00006, two-tailed unpaired Student's t-test). For the boxplot, + represents mean, center line represents median, box represents interquartile range between first and third quartiles, whiskers represent 1.5* interquartile range, and dots represent outliers. ***p < 0.0005.

## Discussion

In this study, we propose that CLN3 is localized to apical microvilli of RPE and is essential for crucial structure (RPE microvilli) and function (POS phagocytosis) of RPE cells that are vital for photoreceptor survival and therefore vision. Using hiPSC-RPE from patients harboring a homozygous 966 bp deletion spanning exon 7 and 8, the most common mutation in CLN3 disease, we show that disease-causing *CLN3* mutations in CLN3 disease affect both RPE cell structure and function in a cell autonomous manner. Specifically, a proportion of CLN3 in human, mouse and hiPSC-RPE cells localized to the RPE microvilli, the RPE structure necessary for POS phagocytosis. Furthermore, disease-causing mutations in *CLN3* led to loss of CLN3 function which subsequently decreased apical RPE microvilli density and reduced POS binding that resulted in lower uptake of POS by CLN3 disease hiPSC-RPE cells. Of note, consistent with the requirement of CLN3 for POS phagocytosis, lower uptake of POS could be rescued by lentivirus-mediated WT-CLN3 overexpression in CLN3 disease hiPSC-RPE cells. Also, consistent with reduced POS uptake, CLN3 disease hiPSC-RPE displayed decreased autofluorescent POS-breakdown products. Notably, these results are in alignment with reduced lipofuscin (autofluorescent POS-breakdown product) observed in CLN3 disease donor eyes[13] (Fig. 2A, B and Supplementary Fig. 2), ERG recordings[20,21] and the retinal imaging pattern[64], (Fig. 9) in early stage CLN3 disease in vivo. Altogether, these results illustrate a role of CLN3 in regulating POS phagocytosis in human RPE cells and (*i*) suggest a role of primary RPE dysfunction in CLN3-associated retinal degeneration and (*ii*) indicate gene-therapy targeting RPE cells as a potential treatment option to suppress photoreceptor cell loss in CLN3 disease caused due to the common 966 bp deletion.

The photoreceptor–RPE interface plays a crucial role in retinal homeostasis and is the primary site of disease pathology in several retinal degenerative diseases (e.g. Best disease[36], retinitis pigmentosa[29]). In fact, both structural pathology affecting the RPE microvilli and functional alterations impacting POS phagocytosis can notably affect the ability of RPE cells to support photoreceptor cell health. For instance, loss of RPE apical microvilli arising from mutations in *EZR* and *SOD2* has been linked to the development of retinal degeneration[65–67]. Similarly, defective

enlarged (Fig. 9H) and POS loss was observed in the area corresponding to previously documented hyperautofluorescent annulus (Fig. 9I). Altogether, longitudinal multimodal imaging on a CLN3 disease patient and previously documented ERG changes in CLN3 disease patients before the loss of vision[20,21] are concordant with POS phagocytosis defect causing photoreceptor cell loss and reduced RPE lipofuscin in CLN3 disease retina. This is critical for therapeutic intervention in CLN3 disease especially when taken in conjunction with our previous data (Fig. 8) showing rescue of POS phagocytosis defect by overexpression of WT CLN3 in CLN3 disease hiPSC-RPE cells.

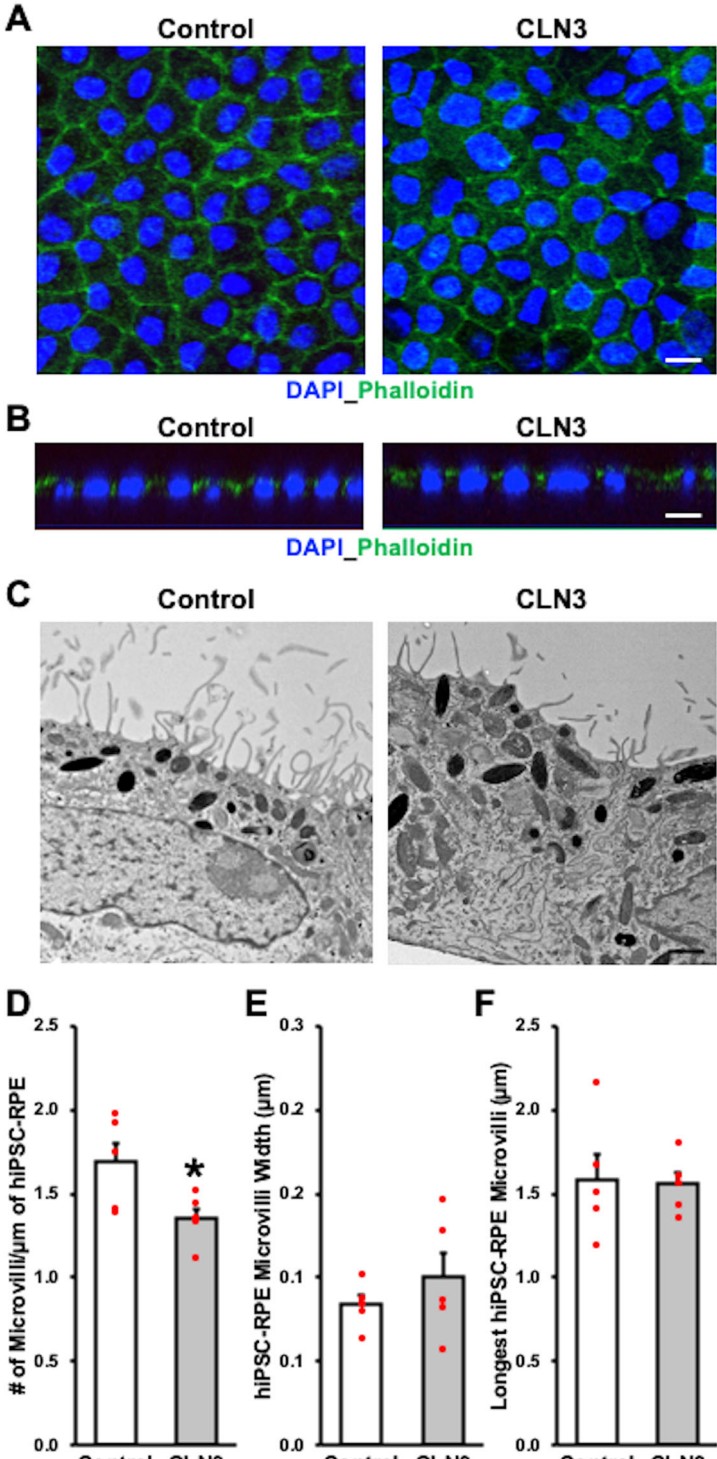

**Fig. 6 Decreased apical RPE microvilli density in CLN3 disease hiPSC-RPE cultures. A, B** Representative confocal microscopy images after immunocytochemical analyses showing similar localization of F-actin as visualized by phalloidin staining in control and CLN3 disease hiPSC-RPE cells (scale bar = 10 μm) ($n$ = 3). Of note in panel **B**, orthogonal view of hiPSC-RPE monolayer shows expected apical localization of phalloidin relative to cell nuclei (DAPI) in control and CLN3 disease hiPSC-RPE. **C–F** Representative transmission electron microscopy (TEM) images (**C**, scale bar = 1 μm) and corresponding quantitative analyses showing decreased apical RPE microvilli density (**D**, $p$ = 0.042), but similar apical-RPE microvilli width (**E**, $p$ = 0.38), and length (**F**, $p$ = 0.88) in control versus CLN3 disease hiPSC-RPE cells ($n$ = 5). Two-tailed unpaired Student's $t$-test performed for all statistical analysis. *$p$ < 0.05.

POS phagocytosis caused by mutations in *MERTK* leads to photoreceptor cell loss and retinal degeneration[29,68]. It is noteworthy that despite reduced POS phagocytosis by CLN3 disease hiPSC-RPE cells, the protein expression of the POS binding receptor, avß5 integrin (Supplementary Fig. 6), the POS engulfment receptor, MERTK (Supplementary Fig. 6), and RPE apical microvilli regulator, EZR (Fig. 1E), were unchanged between control and CLN3 disease hiPSC-RPE cells. CLN3 has been

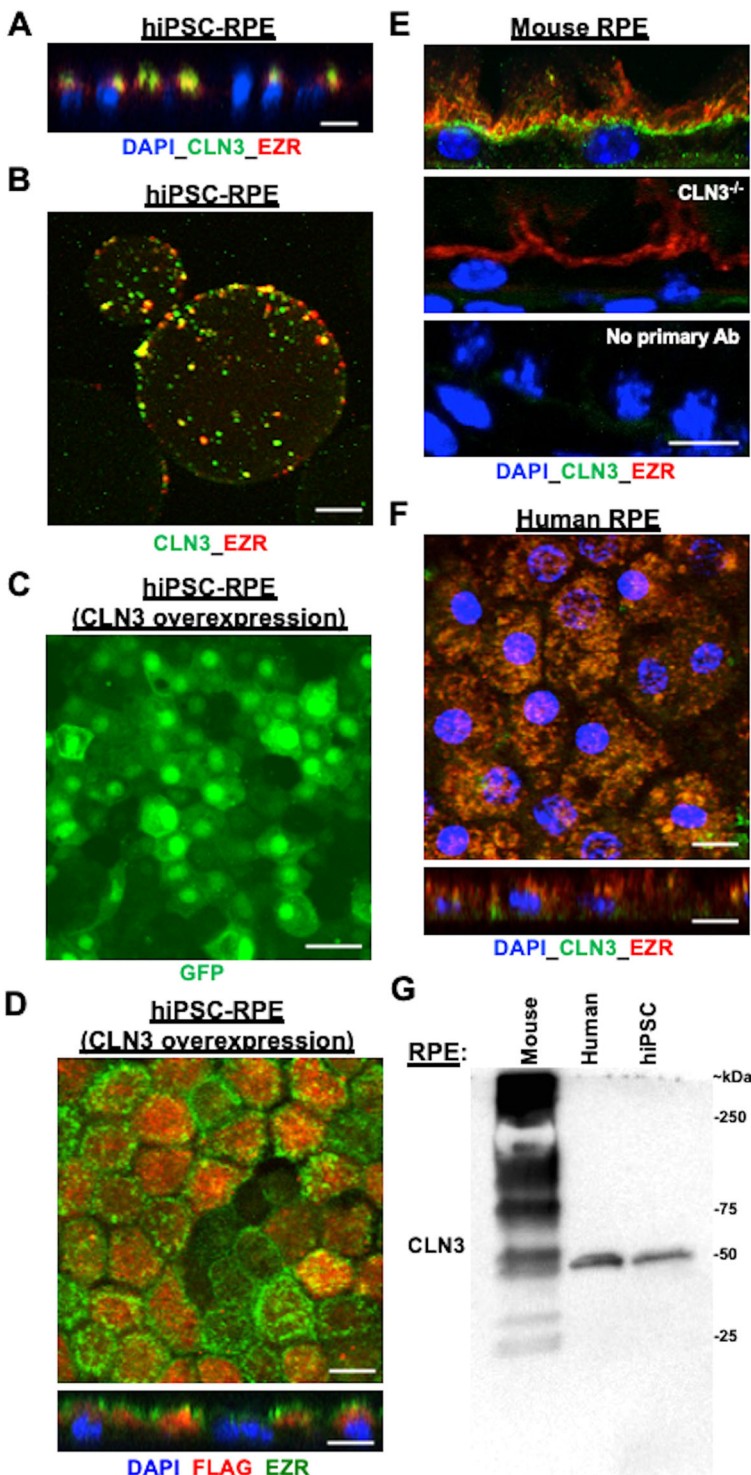

shown to regulate cytoskeletal architecture in other cell types[69,70]. In conjunction with CLN3 (Fig. 7), other proteins involved in cytoskeletal organization, including F-ACTN, are major constituents of apical RPE microvilli[50,71]. However, immunocytochemical localization revealed a similar expression pattern of F-ACTN in control versus CLN3 disease hiPSC-RPE cells (Fig. 6A, B). It is still plausible that reduced apical microvilli density (Fig. 6C, D) directly contributes to decreased POS binding and uptake by CLN3 disease hiPSC-RPE cells. Of note, although there is currently limited information on RPE microvilli formation/turnover, the RPE microvilli comprises of densely packed actin filaments and other cytoskeleton components, including EZR and Ezrin-Radixin-Moesin-Binding Phosphoprotein 50 (EBP50)[72,73]. Notably, lack of F-ACTN and/or EZR can impact microvilli formation[67,74] and thereby impact RPE cell's phagocytic ability by impacting POS binding[46]. Furthermore, cytoskeletal rearrangement is crucial for POS ingestion by RPE cells[75]. Given that CLN3 has been shown to (i) interact with non-muscle myosin IIB (NM-IIB), an F-ACTN binding motor protein that has been suggested to play a role in POS phagocytosis[69] and (ii) impact cytoskeletal reorganization in other cell type(s)[76], it is plausible that CLN3 could impact microvilli formation and turnover as well as POS

**Fig. 7 A proportion of CLN3 protein localizes to RPE microvilli in both (*i*) control hiPSC-RPE cells with and without CLN3 overexpression and (*ii*) primary human and mouse RPE cells. A** Representative confocal microscopy image of the orthogonal view of control hiPSC-RPE monolayer post immunocytochemical analyses with antibodies against CLN3 and EZR (an RPE microvilli protein) showing apical presence of endogenous CLN3 and co-localization of endogenous CLN3 and EZR (scale bar = 10 μm) (n ≥ 3). **B** Representative confocal microscopy image post microvilli isolation with lectin-agarose beads and immunocytochemical analyses with CLN3 and EZR antibody showing co-localization of endogenous CLN3 and EZR in the control hiPSC-RPE microvilli-bound to lectin-agarose beads (scale bar = 10 μm) (n = 3). **C** Representative confocal microscopy images post immunostaining with a GFP antibody showing robust expression of EGFP in control CLN3 hiPSC-RPE cells transduced with pHIV-FLAG-IRES-EGFP lentiviral vector (scale bar = 10 μm) (n ≥ 3). Of note, the observed EGFP localization in the nucleus is due to the nuclear and kinetic entrapment of EGFP homomultimers, and has been previously reported[53]. **D** Representative confocal microscopy images post immunocytochemical analyses with FLAG and CLN3 antibodies showing co-localization of FLAG-CLN3 in control hiPSC-RPE cells transduced with pHIV-FLAG-CLN3-IRES-EGFP lentiviral vector (scale bar = 10 μm) (n ≥ 3). Of note, nuclei were stained with DAPI and are excluded in the top panel showing the **D** image to better visualize the CLN3-EZR co-localization. In the bottom panel showing the orthogonal view, DAPI is included to illustrate the apical localization of both FLAG-CLN3 and EZR. **E** Confocal microscopy analyses of mouse retina sections after immunocytochemical analyses with CLN3 and EZR antibody showed co-localization of endogenous CLN3 with EZR (top panel). Notably, CLN3 antibody fails to detect CLN3 expression but EZR can be visualized in the RPE cells of CLN3$^{-/-}$ mice (**E**, middle panel). Furthermore, no specific CLN3 staining was seen in WT mouse retina sections in the no primary controls that excluded incubation with primary antibody (**E**, bottom panel). Of note, because the host of CLN3 antibody is mouse, we utilized mouse on mouse (M.O.M ®) kit in these experiments (scale bar = 10 μm) (n ≥ 1). **F** Confocal microscopy images post immunocytochemical analyses showed co-localization of endogenous CLN3 and EZR in primary human RPE wholemounts and orthogonal view (scale bar = 10 μm) (n ≥ 1). **G** Representative western blot image showing presence of endogenous CLN3 protein (50 kDa) using CLN3 specific antibody in WT mouse, primary human and hiPSC-RPE samples (n ≥ 1). Also note a distinct pattern of CLN3 in mouse versus human RPE cells. Specifically, multiple bands for CLN3 were seen in native mouse RPE (~15–20 μg total protein) compared to human RPE (primary, hiPSC) possibly due to non-specific background due to the use of mouse CLN3 antibody.

phagocytosis in RPE cells through its interactions with cytoskeletal proteins, such as NM-IIB and F-ACTN. Furthermore, primary dysfunction within RPE microvilli and lack of microvilli has been shown to impact photoreceptor–RPE interaction and lead to vision loss in mice lacking EZR protein[67] and in a canine model of inherited retinal degeneration, Best disease[77]. Another possibility is that the impaired RPE phagocytosis of POS by CLN3 disease is a consequence of impaired endocytosis. For instance, several studies have suggested a crucial role of CLN3 in the endosomal-lysosomal pathway[78–80] and the endolysosomal system is involved in regulation and recycling of plasma membrane components required for POS phagocytosis (e.g. αVβ5 integrin[81]) and microvilli formation[82]. It is also possible that some of the molecular defects in CLN3 disease hiPSC-RPE cells were masked in our study as we compared CLN3 disease hiPSC-RPE cells to control hiPSC-RPE cells constituting of heterozygote carriers (family members) and unrelated healthy subjects. However, we did confirm no difference in POS phagocytic capability and microvilli density between CLN3 heterozygote carriers and healthy subjects with no CLN3 defect. Furthermore, though gene-corrected lines were not included in this study, as the CLN3 mutation investigated here leads to homozygous 966 bp deletion spanning exon 7 and 8[83], making it challenging for genome editing/CRISPR correction, we utilized WT-CLN3 overexpression in CLN3 disease hiPSC-RPE cells to investigate the role of CLN3 in POS phagocytosis. The correction of *CLN3* mutation in CLN3 disease hiPSCs and introduction of *CLN3* mutation in control hiPSC-RPE cells will be valuable tools to further investigate CLN3 function and CLN3 disease pathophysiology. Similarly, a comparison of molecular and structural defects between CLN3 disease hiPSC-RPE cells and hiPSC-RPE derived from patients with CLN3-associated non-syndromic retinal degeneration would be instrumental in further elucidating the role of RPE dysfunction in instigating photoreceptor degeneration in CLN3 disease versus non-syndromic CLN3-associated retinal degeneration. Notably and consistent with an important role of CLN3 in outer retina homeostasis, in the non-syndromic form of retinal degeneration caused by other mutations in *CLN3*, the initial defects in the retina are restricted to the photoreceptor–RPE complex[21,84].

A recent study using CLN3 disease rodent model has implicated impaired lysosomal degradation of POS by RPE cells in

CLN3 disease pathology[31]. This hypothesis is particularly attractive because CLN3 has been postulated to play a critical role in lysosomal homeostasis in multiple cell types[2,54,85,86]. Furthermore, impaired lysosomal degradation of POS membranes by RPE cells has been experimentally linked to loss of photoreceptors in mouse models[18,87]. Consistent with a role of CLN3 in lysosomal function, a proportion of CLN3 in hiPSC-RPE cells co-localized with lysosomal marker, LAMP1 (Supplementary Fig. 9). However, we did not pursue the role of CLN3 in POS degradation in the current study as impaired lysosomal degradation of POS by CLN3 disease hiPSC-RPE cells, while plausible, is in direct contrast to the decreased accumulation of POS digestion products observed in CLN3 disease hiPSC-RPE cells[2] (Fig. 2). In contrast, POS binding and uptake defect by CLN3 disease hiPSC-RPE cells is consistent with both reduced accumulation of lipofuscin in the RPE and increased autofluorescence in photoreceptor cell layer[1,13,14] that could plausibly be POS debris. In fact, as shown here (Fig. 9), a paracentral hyperautofluorescent annulus is seen in CLN3 disease[21] and in other retinal degenerations[88], which may be representative of increased accumulation of POS debris at the transition zone between healthy and unhealthy photoreceptors. Ultimately, if RPE dysfunction and POS phagocytosis defect are central to photoreceptor cell loss and retinal degeneration in CLN3 disease and CLN3-associated non-syndromic retinal degeneration, RPE cells may be a crucial gene therapy target for these blinding disorders.

## Methods

**hiPSC generation.** Fibroblasts from CLN3 disease patients harboring the homozygous 966 bp deletion spanning exon 7 and exon 8[83] and unaffected controls, heterozygote family members, and unrelated healthy subjects with no known history of retinal disease, were reprogrammed to hiPSCs using non-integrating episomal plasmid vectors in accordance with a previously published protocol[37]. Specifically, using the nucleofection kit for primary fibroblast (Lonza), fibroblasts (~60,000) were electroporated with 1 μg each of pCXLE-hOCT4-shP53, pCXLE-hSK, pCXLE-hUL plasmids (Addgene plasmid #27077, 27078, 27080) in a nucleofector 2b device (Lonza, Program T-016). Following electroporation, fibroblasts were cultured in high glucose DMEM containing 10% FBS for 6 days. Subsequently, $1 \times 10^5$ cells were plated onto irradiated mouse embryonic fibroblast (MEF) feeder layer. The next day the medium was switched to hiPSC basal medium (DMEM/F12 with 20% knockout serum replacement or KSR, 1% MEM-NEAA, 1% glutamax and 100 ng/ml FGF2). hiPSC colonies on MEF feeder layer began to appear 17–30 days post-transfection and individual hiPSC colonies were manually dissected and expanded for characterization.

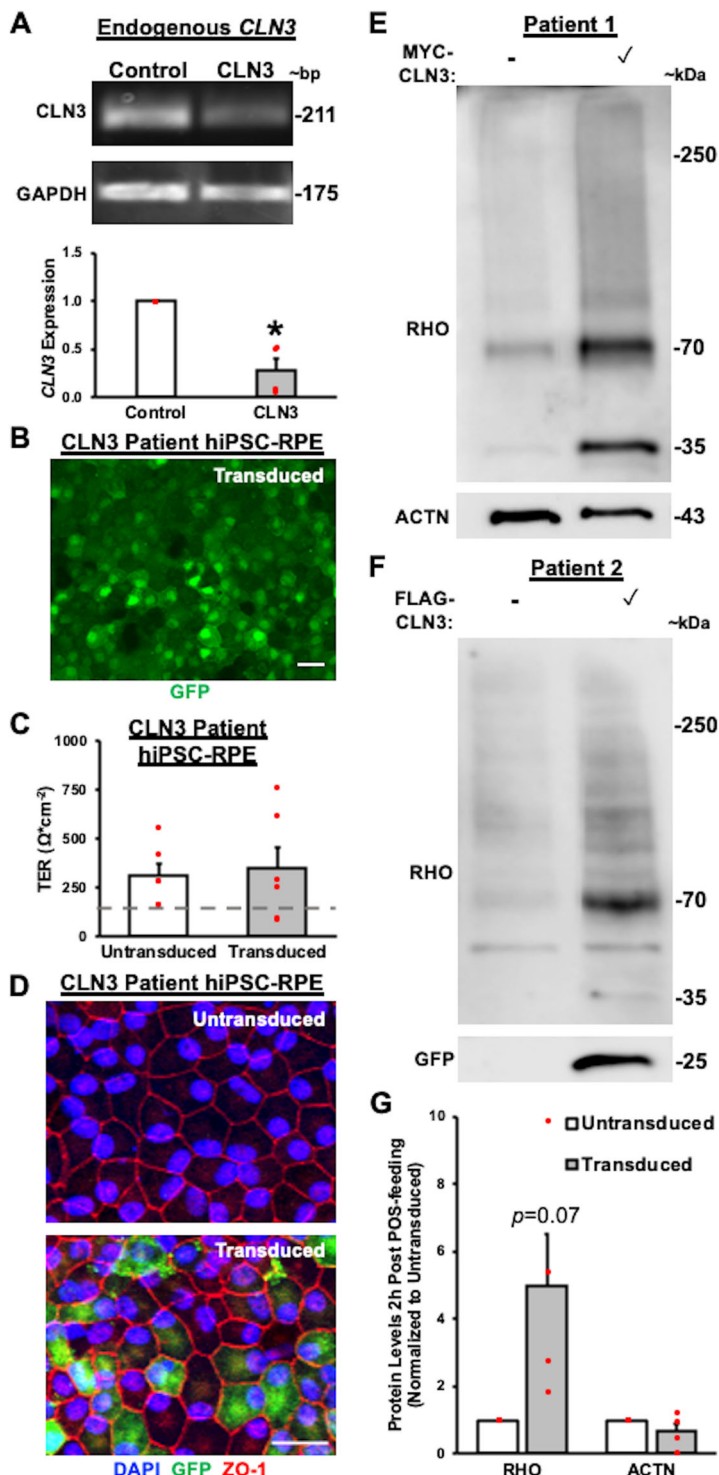

**hiPSC characterization**. Immunocytochemical analyses were performed to confirm the expression of known pluripotency markers, OCT4 and NANOG, in all control and CLN3 disease hiPSC lines (Supplementary Fig. 11). Furthermore, both PCR analyses and gel electrophoresis (Supplementary Fig. 11) and DNA sequencing (Supplementary Fig. 12) using previously published primers (Forward: 5′-CATTCTGTCACCCTTAGAAGCC-3′; Reverse: 5′-GGCTATCAGAGTCCA-GATTCCG-3′)[89] were utilized to confirm the absence and presence of the 966 bp homozygous gene deletion spanning exon 7 and 8 in control versus CLN3 disease hiPSCs respectively. Standard karyotyping analysis was also carried out to confirm chromosomal integrity of all hiPSC lines (Supplementary Fig. 11).

**hiPSC culture and differentiation**. hiPSCs were cultured on either irradiated mouse embryonic fibroblast (Thermo Fisher Scientific) feeder layer in hiPSC basal medium (DMEM/F12 with 20% knockout serum replacement or KSR, 1% MEM-NEAA, 1% GlutaMAX, and 100 ng/ml FGF2 (Pepro Tech) or in feeder-free culture on dishes coated with Matrigel (Corning) in mTeSR1 or mTesR Plus (STEMCELL Technologies). Differentiation to retinal cell fate to RPE was conducted using previously published protocols[37,90]. Summarized here, colonies of hiPSCs cultivated on MEF or mTESR were lifted and cultured in either hiPSC basal medium without FGF or mTESR or mTESR+ to generate embryoid bodies (EBs). Six days post-EB generation, EBs were plated onto laminin-coated tissue culture plates and fed with neural induction medium or NIM (DMEM/F12 containing 1% MEM-NEAA, 1% N2 supplement) and 2 μg/ml heparin. On day 14 post-EB generation, the cell culture medium was switched from NIM to retinal differentiation medium (RDM) containing 70% DMEM/30% F12 and B27 supplement without retinoic acid (RPE differentiation). OV-like structures were either collected on ~day 20 by lifting or dissection and remaining cells were grown as adherent cultures. RPE cells

**Fig. 8 Lentiviral-mediated overexpression of wild-type (WT)-CLN3 increases the amount of POS phagocytosed by CLN3 disease hiPSC-RPE cells.**
**A** Qualitative gel electrophoreses images (top panel) and quantitative real-time PCR analyses (bottom panel) showing reduced expression of endogenous CLN3 gene in control versus CLN3 disease hiPSC-RPE cells (Control: $n = 3$, CLN3: $n = 4$; $p = 0.0051$) normalized to control hiPSC-RPE expression. GAPDH served as loading control. **B** Representative confocal microscopy images post immunostaining with GFP specific antibody showing robust expression of GFP in transduced CLN3 disease hiPSC-RPE cells (pHIV-FLAG-CLN3-IRES-EGFP). Scale bar $= 20$ μm. **C** Transepithelial resistance (TER) measurements showing no adverse effect of transduction on epithelial integrity in CLN3 hiPSC-RPE cells transduced with either pHIV-MYC-CLN3-IRES-EGFP or pHIV-FLAG-CLN3-IRES-EGFP lentiviral vectors (Control: $n = 4$, CLN3: $n = 6$; $p = 0.79$) in both control and CLN3 disease hiPSC-RPE cells grown as a monolayer on Transwell inserts. (Dotted line, TER reported in vivo threshold of ~150 $\Omega\,cm^{-2}$ [42,43]). **D** Representative confocal microscopy images showing similar localization of tight junction protein, ZO-1, in transduced CLN3 disease hiPSC-RPE cells and untransduced CLN3 disease hiPSC-RPE cells. Of note, robust expression of GFP can be seen in CLN3 disease hiPSC-RPE transduced cells (pHIV-FLAG-CLN3-IRES-EGFP). Of note, GFP localization in the nucleus is due to the nuclear and kinetic entrapment of EGFP homomultimers [53]. Scale bar $= 20$ μm, ($n \geq 3$). **E–G** Representative Western blot images (**E**, **F**) post 2 h POS feeding (monomer band 35 kDa, dimer band 70 kDa, and aggregate/multimer bands between 35–70 kDa and >70 kDa) and quantitative analyses (**G**) showing increased amount of RHO relative to total protein ($p = 0.07$), but similar levels of ACTN relative to total protein ($p = 0.24$) in CLN3 disease hiPSC-RPE cells expressing WT-CLN3 (cells transduced with either pHIV-MYC-CLN3-IRES-EGFP or pHIV-FLAG-CLN3-IRES-EGFP lentiviral vectors) compared to untransduced CLN3 disease hiPSC-RPE cells ($n = 4$). Of note, as expected GFP band was observed only in transduced CLN3 disease hiPSC-RPE cells at ~27 kDa due to usage of bi-cistronic lentiviral pHIV-MYC-CLN3-IRES-EGFP and pHIV-FLAG-CLN3-IRES-EGFP vectors (**F**). Two-tailed unpaired Student's $t$-test performed for all statistical analysis. $*p < 0.05$, $**p < 0.005$.

post-dissection were cultured as adherent monolayer in RDM without retinoic acid. To passage RPE cells, 0.05% trypsin was utilized to dissociate the cells. Subsequently, hiPSC-RPE cells were plated onto laminin-coated (4–24 h) 24 well plates and/or transwells and cultured in RDM media with 2% FBS until the formation of confluent RPE monolayer. Thereafter, FBS was removed from the cell culture media and cells were exclusively maintained in RDM. Post- hiPSC-differentiation, hiPSC-OVs were either lifted (~day 20) or dissected (day 35–90) from adherent cultures and maintained in retinal differentiation medium (70% DMEM/30% F12 and B27 supplement; Thermo Fisher Scientific) until utilized in experiments. hiPSC-RPE cells showing their characteristic morphology were dissected at ~60–90 days post-hiPSC-differentiation from adherent cultures [34,36]. The dissected pure patches of RPE (P0) were passaged onto non-permeable plastic support (P1) and subsequently re-passaged (P1, P2) onto non-permeable plastic support and subsequently re-passaged (P2, P3) onto 0.4 μm pore size Transwell inserts (Costar, Corning) [34,37]. Of note, passaging of hiPSC-RPE cells was limited to P3 to avoid epithelial to mesenchymal transition [34].

**Processing of primary and hiPSC-RPE samples for wholemount and sections**. Primary and hiPSC-RPE were fixed in paraformaldehyde or Davidson's fixative (hiPSC-RPE: 4% PFA, 30 min; Mouse RPE: 4% PFA, 90 min; Human RPE: Davidson fixative, 72 h; Human RPE: PFA: 2% long-term). Subsequently, anterior segment was dissected out of mouse and human eyes and retina/RPE in the optic cup was processed for wholemount preparation and sectioning. hiPSC-RPE and primary RPE samples for wholemount analysis were directly utilized for immunocytochemistry (see Immunocytochemical analysis). For processing frozen section, fixed samples were washed and passed through sucrose gradient. Specifically, primary (mouse) eyes were washed 2X in PBS and incubated in 10% sucrose o/n at 4 °C, followed by 20% sucrose o/n at 4 °C, and subsequently 30% sucrose for 3 days at 4 °C. Similarly, hiPSC-RPE bound to transwell membranes were washed 2X in PBS and incubated in sucrose solution, 10% for 1 h, 20% for 1 h and 30% o/n. After processing through sucrose gradient, samples were placed in tissue freezing medium (Triangle Biomedical Sciences), snap frozen and sectioned at 14 μm thickness at −20 °C in a Cryostar NY50 (Thermofisher Scientific). Sectioned slides were either directly used for immunocytochemistry or stored at −20 °C until further use.

**Immunocytochemical analysis**. hiPSC-RPE, human, and mouse eyes were prepared and processed for immunocytochemistry. Immunocytochemistry was performed as previously described [37,91]. Briefly, fixed hiPSC-RPE and primary (mouse, human) RPE wholemounts were blocked/permeabilized in PBS containing 10% normal donkey serum (ImmunoReagents Inc) and 0.1–0.4% triton-X-100 for 1 h and o/n respectively. Subsequently, wholemount samples were incubated in the primary antibody solution in 0.5X blocking buffer at 4 °C for o/n (hiPSC-RPE) or 3 days (mouse) or 4 days (human RPE). Following this, hiPSC-RPE and primary RPE samples were washed 2–4 times in PBS-triton X-100 and incubated in secondary antibody solution in 0.5X blocking buffer for either 1 h at room temperature (hiPSC-RPE) or 2 days at 4 °C (primary RPE). Following another 2 washes in PBS-TX hiPSC-RPE and primary RPE samples were incubated in nuclear staining dyes, DAPI or Hoescht 33342 (life technologies) for 15–30 min. In a subset of experiments, hiPSC-RPE samples were also incubated with Alexa 633-Phalloidin for 30 min. Subsequently, hiPSC-RPE and primary RPE samples were mounted in Prolong gold (Life Technologies), coverslipped and imaged using confocal microscope (LSM 510 META, Zeiss). For frozen mouse RPE sections evaluating CLN3 localization using a mouse antibody, immunocytochemistry was carried out using Vector® M.O.M™ immunodetection kit (Vector laboratories) in accordance with the manufacturer's instructions. Primary antibody used in this study included

BEST1 (1:50; Millipore MAB5466), CLN3 (1:50; Santa Cruz Biotechnology SC-398192), EZR (1:1000, Cell Signaling 3145S), FLAG (1: 500, Sigma F3165), GFP (1:1000, Novus Biologicals NB600-308), MYC (1:8000, Cell Signaling 2276S), OCT3/4 (1:100, Santa Cruz Biotechnology SC-5279), NANOG (1: 100, R & D systems AF1997), LAM (1:200, Abcam ab210956), LAMP1 (1: 200, Abcam ab24170), and ZO-1 (1:100, Life Technologies 61-7300). Alexa-conjugated secondary antibodies (1: 500) were used for all experiments, as well as 633-Phalloidin (Life Technologies A22284). Of note, in a subset of experiments utilizing CLN3 antibody, epitope-specific blocking peptide (Santa Cruz Biotechnology SC-398192P) was used to confirm the specificity of the CLN3 antibody in accordance with the manufacturer's recommendation.

**Western blot analysis**. Total cellular protein was isolated either in RIPA buffer (ThermoFisher Scientific) or Tris-triton (1% triton in 1X TBS) containing protease inhibitor (Sigma). Subsequently, Biorad-DC protein assay was used to quantify total cellular protein and samples were resolved on 4–20% gradient gels by SDS-PAGE. Afterward, gels were transferred onto PVDF membrane as previously described [37] and briefly summarized here. Subsequent to protein transfer, PVDF membranes were incubated in blocking buffer, 5% dry milk in PBS and/or commercially bought blocking buffer (Licor) for 1 h at room temperature and then washed 4 times in 0.1 % PBS-Tween. Next, PVDF membranes were incubated in protein-specific primary antibody in blocking buffer at 4 °C o/n. This was followed by 4 more washes in PBS and incubation in host-specific infrared (Licor) or HRP (Azure) secondary antibody solution in blocking buffer for 1 h at room temperature. After four more PBS washes, the PVDF membrane was visualized and analyzed using Azure C500 imaging system (Azure Biosystem). Primary antibody used in this study for Western blot analyses included ACTN (1:750, Santa Cruz Biotechnology SC-47778), BEST1 (1:500, Millipore MAB5466), CLN3 (1:500 Santa Cruz Biotechnology SC-318192) CRALBP (1: 10,000, Abcam ab15051), EZR (1:1000, Cell Signaling Technology 3145S), FLAG (1:2000, Sigma F1804), MYC (1:1000, Cell Signaling Technology 2276S), GFP (1:1000, Novus Biologicals NB600-308), ITGAV (1:500, Santa Cruz Biotechnology SC-376199), ITGB5 (1: 500, Abcam ab184312), MERTK (1:500, Abcam ab52968), RHO (1:500, Millipore MABN15) and RPE65 (1:500, Millipore MAB5428). Images of all original blots are available in the Supplementary data (Supplementary Fig. 13).

**Quantitative real-time PCR**. RNA from hiPSC-RPE cells was isolated, processed, and analyzed using previously published gene-specific primer sequences [37]. RNA was isolated from hiPSC-RPE cells using the QiaShredder and RNAeasy micro kit (Qiagen, Germantown, MD) and treated with DNase I for 30 min. cDNA was synthesized using the iScript reverse transcriptase kit (BioRad). SYBR Green (BioRad) and gene-specific primers were used along with the synthesized cDNA on the CFX-Connect Real Time System cycler (BioRad) to quantitate using GAPDH as the loading control. Of note, all the primers used in this study are listed in Supplementary Table 1.

**TER measurements**. TER of hiPSC-RPE monolayer in transwell inserts was recorded as previously described [34] using an EVOM2 volt-ohm meter (World Precision Instruments). In every TER measurement session, a blank recording was taken using a cell-free transwell containing RPE cell culture medium alone was carried out. After blank subtraction, TER measurements were reported as resistance per area or $\Omega\cdot cm^{-2}$.

**POS phagocytosis assay**. Bovine POS was obtained commercially from InVision BioResources (Cat. #98740, Seattle, WA) and has previously been utilized by

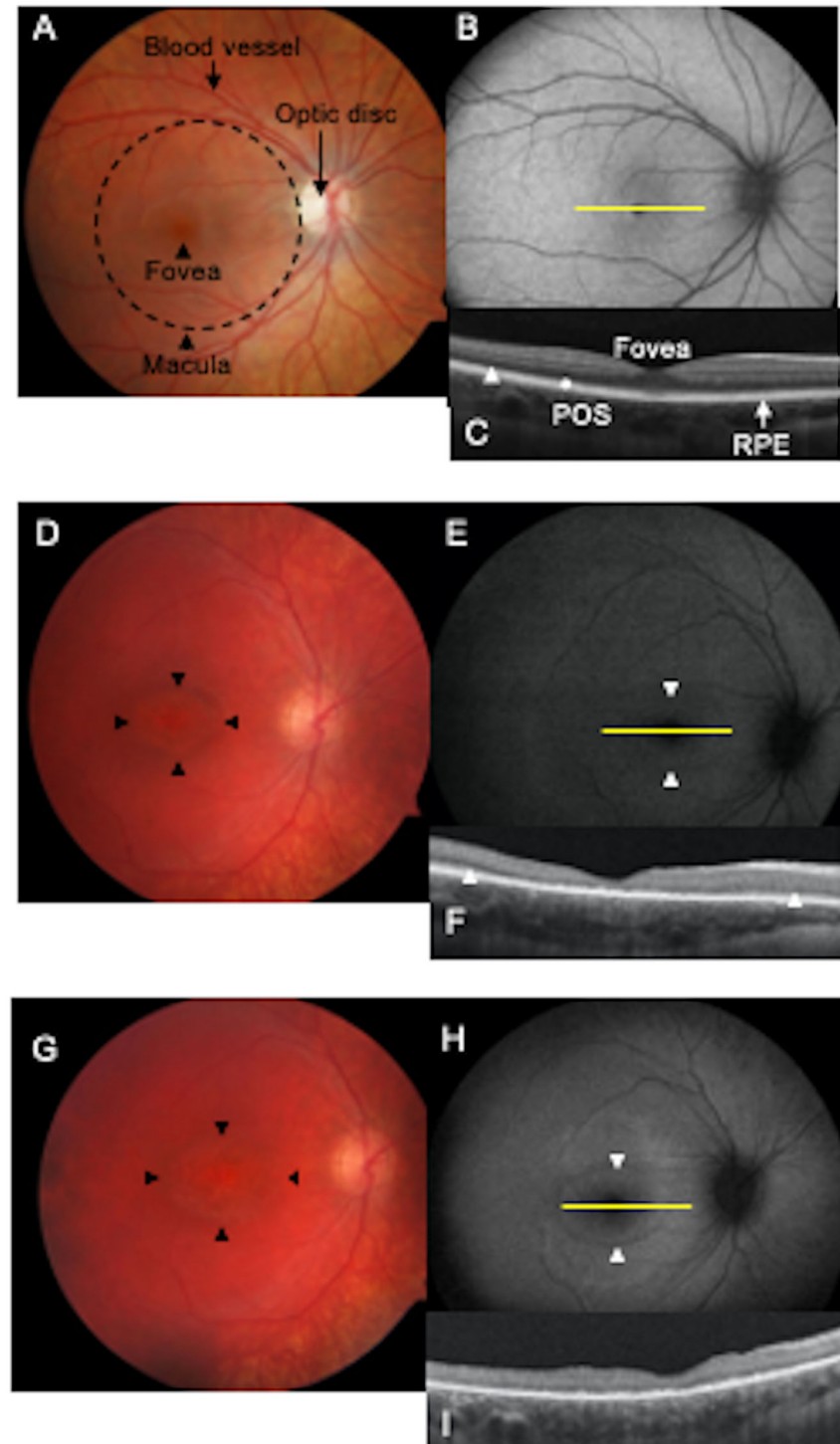

**Fig. 9 Longitudinal multimodal imaging of a CLN3 disease patient is consistent with autofluorescent changes in photoreceptor–RPE complex preceding photoreceptor cell loss. A–C** Multimodal imaging of a normal eye. **A** Fundus photograph of a normal right eye. Dashed circle outlines roughly the macula. The fovea is marked with an arrowhead in the center of the macula. Blood vessel and optic disc are also labeled (yellow lines). **B** Fundus autofluorescence (FAF) image of a normal right eye. Note the optic disc and the blood vessels are dark as they do not exhibit autofluorescence. Optical coherence tomography (OCT) section of the retina through the fovea (yellow line in **B**) is shown. **C** The photoreceptor nuclei are packed in the hypo-reflective band (arrowhead). The retinal pigment epithelium (RPE) is the linear hyper-reflective band marked by a line. Photoreceptor outer segments (POS) are present in the area between the photoreceptor nuclei layer and the RPE as alternating hyper-and hypo-reflective linear bands (marked with an asterisk). **D–I** Multimodal imaging in a 7-year-old patient with CLN3 disease. **D** Fundus photograph showing bull's eye maculopathy (BEM) outlined with arrowheads. **E** FAF showing a faint hyperautofluorescent annulus at the margin of BEM (arrowheads). **F** OCT section through the fovea reveals central loss of the photoreceptor nuclear layer as well as the POS layer. The layers are present towards the periphery on either side of the fovea but the layers are thinner than normal (arrowheads). At follow-up at age 8 years, BEM has enlarged (arrowheads in **G**), hyperautofluorescent annulus has become more diffuse (arrowheads in **H**); and the loss of POS and photoreceptor nuclei can be seen across the entire section of the retina (**I**) with a decrease in overall thickness of the retina as well.

multiple studies to evaluate POS phagocytosis by human primary cells and hiPSC-RPE cells in culture[39,45,75,91]. Parallel cultures of age-matched hiPSC-RPE on transwells were fed either a physiological doses of unlabeled POS or FITC-labeled POS (~20–40 POS/RPE cell)[92] for 30 min to 2 h at either 17 °C (to measure POS binding[47]) or 37 °C (to measure POS binding and internalization). In a subset of experiments, evaluating autofluorescence material accumulation, POS feeding was performed daily for 2 weeks. At the end of the incubation period, cells were washed with 1X PBS to get rid of any unbound POS lying on hiPSC-RPE cell surface[45]. Subsequently, hiPSC-RPE cells were either immediately harvested for Western blot analysis or fixed for immunocytochemical processing. Further details on Western blot processing and immunocytochemical analysis for evaluating POS binding and internalization is as follows: hiPSC-RPE transwell wholemounts post confocal microscopy imaging (FITC-POS, ex: 490 em: 525; Autofluorescence, ex: 546 nm, em: 560–615 nm; ZO-1, ex: 632 em: 647; DAPI, 361, em: 497) were analyzed using Image J (NIH) software. Of note, the areas imaged were chosen randomly by utilizing DAPI staining for cell nuclei, while being blind to FITC-staining and autofluorescence levels. For quantification of autofluorescent material, the count and the area of autofluorescent particles (ex: 546, em: 560-615) per 100 RPE cell nuclei stained with DAPI in parallel cultures of control versus CLN3 disease hiPSC-RPE were evaluated. Similarly, the count of FITC-POS < 5 μm (threshold set to include bound POS and ingested phagosome but exclude POS aggregates)[93] per viewing area was determined in parallel cultures of FITC-POS fed control and CLN3 disease hiPSC-RPE cultures. In a subset of experiments, the number of bound versus internalized FITC-POS were determined using the position of FITC-fluorescence relative to tight junction marker, ZO-1[46]. Briefly, images were taken as Z-stacks using the apical tight junction marker ZO-1 as a guideline for cell orientation (apical versus basal). The Z-stack images were then examined slice-by-slice to decide where the ZO-1 staining was the most intensely labeled, compiling the slices above it to be the "apical" image of the RPE cell and compiling the slices below to be the "basal" image of the RPE. Maximum intensity projections of these separated stacks were used to quantify the number and area of FITC-POS particles apical (bound) vs basal (internalized) of ZO-1 staining within the cell. Of note, for quantitative analysis, at-least five different confocal images (distinct viewing areas) were utilized for each individual sample in an experiment.

**Electron microscopy analysis**. hiPSC-RPE monolayer grown on Transwells were fixed in glutaraldehyde (2.5%) and paraformaldehyde (4%) in sodium cacodylate (0.1 M) and embedded in epoxy resin and 60 nm thick sections were cut in 10 μm depth progression into the sample. Thereafter, hiPSC-RPE sections were imaged using a transmission electron microscope (H-7650 Transmission Electron Microscope; Hitachi) at the University of Rochester Electron Microscopy Core. For assessment of microvilli width, length, and density, TEM images showing cells in the longitudinal section were analyzed using Image J software (NIH) and Microsoft Excel. All measurements were done manually. Microvilli width was measured from the outer membrane to outer membrane, about halfway up on 10 microvilli throughout a single image. On each image the length of the longest microvilli was evaluated, while the microvilli density was calculated by measuring the length of the RPE surface in each section and counting the number of microvilli. Using these measurements, the average number of microvilli per μm of RPE length was determined. A total of about 100 images were analyzed this way, with 10 images from 5 different samples each for control and CLN3 disease hiPSC RPE cultures, respectively.

**Microvilli isolation**. The microvilli of hiPSC-RPE cells were isolated using wheat germ agglutinin (WGA)-beads (Sigma L1394) in accordance with a published protocol[50,94]. Briefly, mature monolayers of RPE cells were treated with WGA-beads in 1X Tris-buffered saline (TBS) for 2–3 h at 4 °C. After the 2–3-h incubation, a syringe was used to spray 1X TBS on RPE cells and dislodge the beads. Subsequently, beads were collected in 1X TBS and processed for Western blot analysis or immunohistochemistry. Next, RPE cells (after bead removal) were scraped off the underlying membrane and collected in 1X TBS buffer and protein was isolated in RIPA buffer for Western blot analysis. Of note, for Western blot analysis, the microvilli-containing WGA-beads were boiled in 2X Laemmli buffer for 15 min. For immunohistochemistry analysis, WGA-beads beads were fixed in 4% PFA for 15 min.

**Generation of pHIV-FLAG-CLN3-IRES-EGFP and pHIV-FLAG-CLN3-IRES-EGFP lentiviral construct**. The lentiviral vector pHIV-FLAG-CLN3-IRES EGFP was generated by cloning FLAG-CLN3 and MYC-CLN3 into the MCS of pHIV-IRES-EGFP vector (Addgene 21373). The open reading frame of human CLN3 was amplified by PCR primer set i) HpaI-ATG-FLAG-CLN3 Forward (TAAGCAGTTAACATGGATTACAAGGATGACGACGATAAGG-GAGGCTGTGCA-GGCTCGCGGCGGCGCTTT) and BamHI-CLN3 Reverse (TGCTTAGGATCCTCAGGA-GAGCTGGCAGAGGAAGTCATGCAG) or ii) HpaI-ATG-MYC-CLN3 Forward (GAACAAAAACTCATCTCAGAAGAGGATCTGGGAGGCTGTGCAGGCTCG CGGCGGCGCTTT) and BamHI-CLN3 Reverse (TGCTTAGGATCCTCAGGAG AGCTGGCAGAGGAAGTCATGCAG). The amplified PCR product was digested using HpaI and BamHI restriction enzymes and ligated into HpaI and BamHI site of pHIV-IRES-EGFP vector to generate bi-cistronic pHIV-FLAG-CLN3-IRES-EGFP or pHIV-MYC-CLN3-IRES-EGFP vector, which allowed simultaneous expression of either i) FLAG-CLN3 and EGFP or ii) FLAG-CLN3 and EGFP

separately, but from the same RNA transcript[51,52]. The ligated plasmids were transformed into XL-1 Blue Competent Cells (Agilent). The plasmid construct was confirmed by restriction enzyme digestion and DNA sequencing.

**Statistics and reproducibility**. Unless stated otherwise, two distinct CLN3 disease iPSC lines both harboring the common 966 bp deletion in the CLN3 gene and three distinct control hiPSC lines including familial heterozygote control and healthy unaffected control that did not harbor the mutation in the CLN3 gene were used for all experiments. Additionally, throughout the study, experiments utilized two different clones from each of these hiPSC lines. Importantly, data presented in the manuscript arises from results that have been consistent across individual clones of each patient line and between the two patient lines. Furthermore, for all experiments with the exception of immunocytochemistry for RPE markers, parallel age-matched cultures of control and CLN3 disease hiPSC-RPE monolayer grown in Transwell inserts with TER > 150 Ω cm$^{-2}$ (reported in vivo threshold[42,43]) were utilized. For immunocytochemical analyses of RPE markers, hiPSC-RPE on both transwells and coverslips were used. For each experiment, data from all control lines and CLN3 disease lines were grouped together and used in control versus CLN3 disease analyses. Data throughout the manuscript is presented as mean ± sem. Significance was computed using two-tailed unpaired student's t-test in Microsoft excel and graphs and charts were plotted using either Microsoft excel and Prism-Graphpad software.

Of note, n in each individual figure refers to the number of biological replicates where data was obtained from hiPSC-RPE cells derived from distinct EB differentiations. Raw data from technical replicates from the same biological replicate are shown in the Supplementary file.

**Human subjects**. Patient sample collection and subsequent experimental analyses were performed in accordance with an approved Institutional Regulatory Board (IRB) Protocol (RSRB00056538) at University of Rochester and conformed to the requirements of the National Institutes of Health and Declaration of Helsinki. Furthermore, for multimodal retinal imaging studies (RSRB00004080) written informed consent was received from the participant/guardian prior to inclusion in the study.

**Procurement of human and mouse tissue**. Whole eyes from healthy adult without any history of retinal disease were obtained from Minnesota Lions Eye Bank, Lions Gift of Sight (time of enucleation: 10.2 h). Retina/RPE from a 22-year-old donor with genotype-phenotype confirmed CLN3 disease was acquired from Dr. Vera Bonilha at Cleveland Clinic (#591; time of enucleation: 4 h). Whole eyes from wild-type and $CLN3^{-/-}$ ($Cln3^{\Delta ex1-6}$) mice were from 5–6-month-old male animals raised at Sanford research. All mice were cared for in accordance to animal protocols approved by Sanford Research's IACUC and in accordance with guidelines set forth by the NIH and AAALAC.

**Cell lines**. hiPSCs used in this study were derived in our laboratory as described here and in prior studies[37,45,91]. 293FT cells were obtained from Life Technologies.

**Reporting summary**. Further information on research design is available in the Nature Research Reporting Summary linked to this article.

## Data availability
Source data for main and supplementary figures are available in Supplementary Data 1. Inquiry of any additional data should be requested to the corresponding author.

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

## Acknowledgements

The authors would like to acknowledge Dr. Katherine Sims and Dr. Susan Cotman at Massachusetts General Hospital, Harvard Medical School, for access to patient and control fibroblast samples. Donor eyes were obtained through the Foundation Fighting Blindness (FFB) Eye Donor Program. Electron microscopy was completed at the University of Rochester Medical Center Electron Microscope Shared Resource Laboratory. This work was supported by the BrightFocus Foundation Macular Degeneration Grant (to R.S.), David Bryant Trust (to R.S.), Foundation of Fighting Blindness Individual Investigator Award (to R.S), Knights Templar eye foundation grant (to R.S), Retina Research Foundation and Research to Prevent Blindness, RPB's Career Development Award (to R.S.), Unrestricted Challenge Grant to Department of Ophthalmology at University of Rochester and the Department of Ophthalmic Research at Cleveland Clinic, and National Institutes of Health grants, R01EY028167 (to R.S), R01EY030183 (to R.S), R01EY027750 (to V.L.B.), P30EY025585 (to CCF), R01NS082283 (to J.W) R01EY027442 (to D.S.W) and P30EY000331 (to D.S.W.).

## Author contributions

C.T., D.S.W., M.C. and R.S. designed research; C.A.G., C.M., C.T. C.S., J.H., K.M., L.W., M.C., M.R., S.D., S.V. and W.S. performed the research and, in addition to D.S.W., V.G. and R.S. analyzed data; E.F.A., J.W.M., J.W., L.L., M.C., T.B.J, V.B. and R.S. contributed new reagents/analytic tools and acquisition of patient and patient samples and mouse tissue; R.S. wrote the paper and all authors read, edited and approved the final manuscript. Of note, in addition to authors C.T. and J.H. that contributed equally to the manuscript, authors K.M. and S.D. also contributed equally to the manuscript.

## Competing interests

The authors declare no competing interests.
