## [Peer Review File · Communications Biology]

Reviewers' Comments:

Reviewer #1:

Remarks to the Author:

This report examines the mechanism whereby a natural deletion mutation of CLN3 lead to retinal degeneration and identify it as a target for gene therapy to treat Batten's disease. CLN3 has been identified as a lysosomal protein. Although the Authors have localized a pool of CLD3 to lysosomes, they have chosen to study a novel function for CLN3 due to its localization to the apical membrane of the retinal pigment epithelium (RPE). A disease-in-a-dish model was created using hiPSC generated from patients and controls that were then differentiated into RPE. This model was combined with studies of mice, primary RPE cultures and cadaveric eyes. The authors draw a connection between the apical pool of CLD3 and phagocytosis, a major function of RPE. Photoreceptors shed the tips of their outer segments on a daily basis, and these "POS" are phagocytized by the RPE. While CLN3 conceivably interacts with known proteins of the phagocytic pathway, the authors do not make this claim and offer an alternative hypothesis. They observe that the mutation is associated with a low density of microvilli and decreased binding of photoreceptor outer segments (POS). A variety of methods are used to corroborate their conclusion that CLD3 is required, directly or indirectly, for phagocytosis, and that expression of wild-type CLN3 restores function. Appropriate statistical analysis is provided, and sufficient detail is provided to reproduce the work. The study should influence future studies of phagocytosis in RPE, and more broadly, those who study phagocytosis and lysosomal function in macrophage and other phagocytic cells. This solid study can be improved by attending to the following details:

1) Besides a simple decrease in the area of the apical membrane (relatively sparse microvilli), the authors might consider impaired endocytosis and recycling of the components needed for binding and uptake of POS or the formation of microvilli. There is two-way intracellular traffic between the plasma membrane, early endosomes, late endosomes, and lysosomes. These pathways regulate recycling, internal storage, and degradation of plasma membrane components. CLDN3 might interfere with some part of this process and link its phagocytic and lysosomal functions. Do the Authors have any data that speak to this possibility? Alternatively, it should be considered in the Discussion.

2) The Authors mention that cadaveric eyes show an accumulation of autofluorescent material and hypothesize that this is accumulated debris. I'm not seeing that data. Reference (2) suggests that is true, but ref 54 within ref 2, says otherwise. Are there any electron micrographs in the literature that support the conclusion of accumulated debris? Since you have the cadaveric eyes, you should examine the photoreceptor layer by fluorescence and EM.

3) Figure 9 would benefit from a more generous use of arrows/arrowheads, and the inclusion of an age-matched normal eye. The optic nerve, blood vessels and macula should be indicated. As it stands, only retinal specialists will be able to interpret the images and understand the description in the text. It is unlikely that this figure will be meaningful for the general Audience of this Journal.

4) First paragraph of the results: Please check reference 45. I do not believe it addresses the TER of RPE in vivo. Two other references would be better to cite here: PMID 1334477 and 24731966.

Lawrence Rizzolo
Professor of Surgery and Ophthalmology
Yale University

Reviewer #2:

Remarks to the Author:

In the manuscript "A human model of Batten disease discloses novel role of CLN3 at the

photoreceptor-RPE interface" Tang et al have studied the role of CLN3 in retinal degeneration. The authors have used retinal pigment epithelium (RPE) cells as a model system to study the role of CLN3 in retinal degeneration. The RPE cells are involved in photoreceptor outer segment (POS) phagocytosis to maintain the health of the photoreceptors. The authors here use Cln3 knockout mice and control and CLN3 patient iPS derived RPE cells as a model system. Overall, this appears to be a comprehensive study of the RPE in the new model they have created, and the results offer some intriguing new insights into mechanisms of retinal degeneration in CLN3 disease. This study should be of significant interest to the field and to investigators studying retinal disease more widely.

The authors have thoroughly characterized the iPS cells before differentiating these into RPE cells. Similar to what is seen in CLN3 patients, the authors show reduced autofluorescence in the CLN3 patient derived cells as compared to the control cells. Further, the authors functionally examined POS uptake and demonstrated decreased POS phagocytosis and that there is a defect in binding of POS in CLN3 deficient cells, which supports a novel hypothesis regarding the role of RPE in the retinal degeneration seen in CLN3 patients and would explain the reduced autofluorescence that is seen in CLN3 patient RPE. Finally, the authors suggest that CLN3 is present in RPE microvilli and have shown that overexpression of CLN3 rescues the phagocytosis defect.

Major comments:

1. The CLN3 antibody data are not convincing and lack several key controls. As shown by Ezaki et al., 2003, J Neurochem, v87, 1296-1308, the CLN3 protein is highly glycosylated and this is reflected in its mobility on SDS-PAGE, showing up as a broad band between ~50 and 60 kDa, depending on the tissue source, due to differential glycosylation. Unfortunately, the authors have not shown any molecular weight standards on their blots throughout the manuscript, so the reader cannot see where the band shown as 'CLN3' is running. More importantly, the band shown is tight and not consistent with the above mentioned, published data. While the authors performed peptide preadsorption controls, the key controls of 1) blot data on lysates prepared from the Cln3 knockout mouse, showing the absence of the band believed to represent CLN3, and 2) blot data on the overexpressed protein, to show that indeed the antibody labels CLN3, are absent. The immunofluorescence data shown in Figure 7E show micrograph images from RPE from a wildtype and Cln3 ko mouse, but given the high level of autofluorescence in RPE, which is reduced when CLN3 is absent, these data too are not convincing without key controls such as no primary controls.
2. Fluorescence data throughout the manuscript are problematic in that there is frequent saturation of signal due to overexposure (e.g. Figure 7, Figure S7). In Figure S7, the colocalization with Lamp 1 and CLN3 are not convincing, also with FLAG-CLN3 and EZR in Figure 7, largely due to the overexposure/saturation of signal. The authors should carefully review the fluorescence images throughout the manuscript and provide non-saturated/non-overexposed data where overexposure is an issue.
3. Detail in the Figure legends are often lacking. For example, in Figure S7, are cells transduced or is endogenous CLN3 stained? If the latter, why is immunoreactivity not homogeneous throughout the cell layer? What antibody is used for the staining if transduced cells; is it FLAG or CLN3 antibody? Lamp1 staining has no characteristic punctuated pattern (most likely due to oversaturated?). Fluorescence in green channel overexposed. Lamp1 staining in CLN3 disease hiPSC-RPE cells in comparison would also be informative. The authors should clarify and should carefully review Figure legends throughout to ensure sufficient detail are provided to the reader to understand the experiments and the data. See additional comments below noting figure legends that require further detail.

Additional comments:

4. Antibody details (catalog numbers) should be provided in the Methods.
5. RHO should be defined when it first appears in the manuscript.
6. Figure 8: Figure legend not comprehensive enough. Figure cannot be understood without having

the main text at hand.

7. Fig 8A: Is endogenous CLN3 measured or transduced CLN3?
8. Fig 8 C, D: Scale bar would be beneficial to compare both images with each other. GFP-expression in transduced cells in B looks cytosolic, whereas it does look clustered in panel D. GFP expression is expected to be cytosolic.
9. Fig 8 D, E: Which cells are patient RPE cells, which are from controls? (Do not understand the sentence 'the amount of RHO is still lower in CLN3 transduced hiPSC-RPE cells when compared to control'. Does it correlate with what is shown in the figure?)
10. Fig 8 D, E: CLN3 or GFP WB band could be shown in addition to rhodopsin in the same blot/ for same samples to show equal transduction of CLN3 disease cells vs. controls
11. Is POS isolated from WT? Was this experiment also tried with POS with CLN3-deficiency? This would add value to the study and strengthen the hypothesis regarding mechanism of retinal degeneration. Additional detail are also required in the Methods for this.
12. Abbreviation for EZR is not stated
13. Figure 1, Panel F: Twice labeled (Ff)
14. Primer used for RT-PCR experiments not listed (in particular those used for CLN3)
15. Authors could explain more specifically how Microvilli length, width was analysed. Manually or automated?
16. In the Discussion, the statement "...we utilized WT-CLN3 overexpression in CLN3 disease hiPSC-RPE cells to confirm a specific and direct role of CLN3 in POS phagocytosis" is an overinterpretation of what was shown. A direct role, particularly given the problems with the data on the CLN3 antibody, has not been shown. Therefore, it is suggested to revise this statement.
17. In the Discussion, because the CLN3 antibody and localization data are unconvincing, the following statement should also be revised "In this study, we show that CLN3 is localized to apical microvilli of RPE and is essential for crucial structure (RPE microvilli) and function (POS phagocytosis) of RPE cells that are vital for photoreceptor survival and therefore vision."
18. In the Discussion, it would be useful for the reader and would strengthen the overall manuscript if the authors included in the Discussion what is known about how RPE apical microvilli are formed/turned over and how this might relate to what is known about CLN3 function in the literature.

Reviewer #3:

Remarks to the Author:

Mutations in CLN3 can lead to loss of photoreceptors. Overall, this is an extensive piece of work with novel and interesting insights. This manuscript reports that CLN3 is required for phagocytosis of photoreceptor (PR) outer segments (POS) by retinal pigment epithelium (RPE) cells, which is essential for PR survival. Patient derived iPSC-RPE cells show defects in microvilli density and reduced POS binding and ingestion.

They report some CLN3 in RPE cells is localised to RPE microvilli, where phagocytosis occurs. Interesting, and relevant to therapeutic development, the phagocytic defects of RPE cells can be rescued by supplying CLN3 gene.

The paper ends with longitudinal imaging of the retina in a CLN3 disease patient which suggests that the autofluorescent changes and a POS phagocytosis defect may precede photoreceptor cell loss in CLN3 disease.

An interesting question from the therapeutic perspective is whether targeting RPE alone is sufficient to rescue loss of PRs, or if targeting PRs, do the RPE also have to be targeted?

Copious data is presented in a clear and logical manner.

I have some comments from a disease perspective:

CLN3-Batten is a new and therefore unusual terminology for this disease – I recommend using one that is more widely recognised.

Juvenile CLN3 disease is the most common type of NCL (as discussed later, other rare types of CLN3 disease may not lose vision, or not show other symptoms besides visual loss).

M&M

What CLN3 antibody was used? How was it derived? What part of CLN3 does it recognise (N terminus, amino acids 13-40)? (See Fig S7). Part of M&M implies it is from Santa Cruz Technology, but this is not clear when all Abs are first listed together.

CLN3 is shown to colocalises with lamp1 in hiPSC-RPE cells cells (Fig S7). Has the same experiment been done in these same cells? Ie does it also colocalise with lamp1? (Could some be mislocalised when over expressed?)

Fig 7 uses eyes from a mouse model that is no longer commonly used (Katz et al 1999), and certainly whose deletion does not match the patient derived cells used – why use this one? How old were the mouse from which the eyes were taken? At what point in the disease?

Figures:

It is not clear which cells were used – M&M suggest there were 3 controls and 2 patient fibroblast lines used to derive iPSC. Presumably representative lines are shown in eg fig 1, 2, 3 and more. How are these chosen? Are the results consistent across all individual lines and clones, and controls? This is important given the genetic variation between lines beyond whether they have a mutation in CLN3.

Important to emphasise that what is observed is specific to the consequences of the 966b deletion of CLN3. Another mutation of the CLN3 gene may give a different response.

Western blots:

In previous research papers, both cytoskeleton and glycolysis have been shown to be affected by the 1kb mutation and loss of CLN3. Thus there is concern about using ACTIN or GAPDH as a suitable normalisation marker. It is suggested that there was no difference in phalloidin staining but it looks more disperse compared to the control, plus it wasn't quantified. Is it possible to normalise to the whole protein level in the lysate to give a more accurate representation. Or show no changes in the same cell line when CLN3 is KD, removed by CRISPR or upregulated. Also bands look pixelated – why is this (compare ACTIN RHO WB in fig4B and fig4 D, fig 7B with EZR).

Minor:

Significance – the 2nd sentence is not correct grammatically.

Reviewer #4: (Trainee - co-reviewed with reviewer #2)

None

Reviewer #5: (Trainee - co-reviewed with reviewer #2)

None

August 18, 2020

Dear Reviewers:

Thank you for providing us the opportunity to submit a revised version of our manuscript **A human model of Batten disease discloses novel role of CLN3 at the photoreceptor-RPE interface.** Your insightful comments and recommendations have allowed us to significantly improve the original manuscript. As you will see from our responses below and the revised manuscript, we have tried to address every single concern raised.

Also, based on the submission system guidelines, text changes in the manuscript and figure legends are marked in red in the "Revised Manuscript- Marked up" file but not the Article file. Furthermore, unless specifically mentioned, the figure numbers in the responses refer to the revised manuscript. Also, text changes in the manuscript that are quoted in the revised response are highlighted in red. In addition, references within the text quoted from the manuscript correspond to the revised manuscript and not the rebuttal letter. Furthermore, new/updated figures along with the legend are presented in the appendix of the rebuttal letter and follow the sequence of reviewer comments in the rebuttal letter. Lastly, we have also included high resolution tiffs for main and supplemental figures to answer questions that may have arisen due to image quality compression in the compiled pdf file

Response to specific reviewer comments

Reviewer 1

Overall comment:

This report examines the mechanism whereby a natural deletion mutation of CLN3 lead to retinal degeneration and identify it as a target for gene therapy to treat Batten's disease. CLN3 has been identified as a lysosomal protein. Although the Authors have localized a pool of CLN3 to lysosomes, they have chosen to study a novel function for CLN3 due to its localization to the apical membrane of the retinal pigment epithelium (RPE). A disease-in-a-dish model was created using hiPSC generated from patients and controls that were then differentiated into RPE. This model was combined with studies of mice, primary RPE cultures and cadaveric eyes. The authors draw a connection between the apical pool of CLN3 and phagocytosis, a major function of RPE. Photoreceptors shed the tips of their outer segments on a daily basis, and these "POS" are phagocytized by the RPE. While CLN3 conceivably interacts with known proteins of the phagocytic pathway, the authors do not make this claim and offer an alternative hypothesis. They observe that the mutation is associated with a low density of microvilli and decreased binding of photoreceptor outer segments (POS). A variety of methods are used to corroborate their conclusion that CLN3 is required, directly or indirectly, for phagocytosis, and that expression of wild-type CLN3 restores function. Appropriate statistical analysis is provided, and sufficient detail is provided to reproduce the work. The study should influence future studies of phagocytosis in RPE, and more broadly, those who study phagocytosis and lysosomal function in macrophage and other phagocytic cells. This solid study can be improved by attending to the following details:

We appreciate the reviewer's enthusiasm and appreciation of the study. As detailed below, we have incorporated all of the reviewer's valuable suggestions.

Specific Comments

1. Besides a simple decrease in the area of the apical membrane (relatively sparse microvilli), the authors might consider impaired endocytosis and recycling of the components needed for binding and uptake of POS or the formation of microvilli. There is two-way intracellular traffic between the plasma membrane, early endosomes, late endosomes, and lysosomes. These pathways regulate recycling, internal storage, and degradation of plasma membrane components. CLN3 might interfere with some part of this process and link its phagocytic and lysosomal functions. Do the Authors have any data that speak to this possibility? Alternatively, it should be considered in the Discussion.

We completely agree with the reviewer and although evaluation of endocytosis/lysosomal functions linked to CLN3 are beyond the scope of the current study, based on the reviewer's valuable suggestion, we have considered this possibility in the revised discussion to state: "Another possibility is that the impaired RPE phagocytosis of POS by CLN3 disease is a consequence of impaired endocytosis. For instance, several studies have suggested a crucial role of CLN3 in the endosomal-lysosomal pathway (64, 85, 86) and the endolysosomal system is involved in regulation and recycling of plasma membrane components required for POS phagocytosis (e.g. $\alpha V\beta 5$ integrin (87)) and microvilli formation (88)."

2. The Authors mention that cadaveric eyes show an accumulation of autofluorescent material and hypothesize that this is accumulated debris. I'm not seeing that data. Reference (2) suggests that is true, but ref 54 within ref 2, says otherwise. Are there any electron micrographs in the literature that support the conclusion of accumulated debris? Since you have the cadaveric eyes, you should examine the photoreceptor layer by fluorescence and EM.

We agree with the reviewer that there is limited literature on evaluating the autofluorescence material in CLN3 disease patient cadaveric eyes due to the rarity of the disease. However, a prior publication (PMID: 19539834) from University of Rochester that included fluorescence analysis of the CLN3 disease cadaver eye by the clinical collaborator on the current manuscript (Dr. Mina Chung) also showed increased autofluorescence material/debris accumulation in the photoreceptor layer. In addition, following the reviewer's recommendation, we have now included representative images of the retina section from the age-matched control and CLN3 disease donor eyes that show increased autofluorescence material/debris in the photoreceptor layer and decrease RPE lipofuscin in the RPE layer of the CLN3 disease donor retina (Fig. 2A). We have also edited the text in the results section to state: "Of note, autofluorescence accumulation was analyzed in the spectral range that is consistent with lipofuscin accumulation in human retina/RPE cells (25, 49). Specifically, confocal microscopy analyses of cryosections and RPE wholemounts of mid-periphery fragments of retina-RPE-choroid showed increased autofluorescence material/debris accumulation in the photoreceptor layer and reduced autofluorescence in RPE cells from CLN3 disease donor eyes compared to RPE autofluorescence in non-CLN3 donor eyes, an age-matched donor eye with Charcot Marie Tooth Disease and adult healthy control donor eye (Fig. 2A, 2B Fig. S2)." Of note, Fig. 2A that was added to the manuscript to address this concern also included in the appendix of this document.

3. Figure 9 would benefit from a more generous use of arrows/arrowheads, and the inclusion of an age-matched normal eye. The optic nerve, blood vessels and macula should be indicated. As it

stands, only retinal specialists will be able to interpret the images and understand the description in the text. It is unlikely that this figure will be meaningful for the general Audience of this Journal.

We agree with the reviewer and have now included fundus, OCT and FAF images from a normal eye in revised Fig. 9 that contains the addition information from normal eyes in panel 9A-C and is annotated with arrow/arrowheads (revised Fig. 9 is included in the appendix of this document. Also, we annotated Fig. 9 with arrow/arrowheads. In addition, we have expanded the figure legend to further describe the data presented in the figure to make it more understandable to general audience of the journal. The revised Fig.9 with the figure legend is also included in the appendix of this document.

Of note, because the reviewer asked for an age-matched normal eye, we would like to emphasize that fundus/FAF/OCT imaging is not done on kids without vision issues and given that the normal retina images shown here are from an individual that is 19 years old and hyperautofluorescence changes (if seen increase with age) and photoreceptor/ POS loss are not seen in normal retina, this image serves as a suitable reference to illustrate the point of POS loss and autofluorescence changes in the Batten disease patient retina.

4. First paragraph of the results: Please check reference 45. I do not believe it addresses the TER of RPE in vivo. Two other references would be better to cite here: PMID 1334477 and 24731966.

Based on the reviewer's recommendation, we have replaced reference 45 with the references suggested by the reviewer (ref 47, ref 48 in the revised manuscript).

Reviewer 2:

Overall comment:

In the manuscript "A human model of Batten disease discloses novel role of CLN3 at the photoreceptor-RPE interface" Tang et al have studied the role of CLN3 in retinal degeneration. The authors have used retinal pigment epithelium (RPE) cells as a model system to study the role of CLN3 in retinal degeneration. The RPE cells are involved in photoreceptor outer segment (POS) phagocytosis to maintain the health of the photoreceptors. The authors here use Cln3 knockout mice and control and CLN3 patient iPS derived RPE cells as a model system. Overall, this appears to be a comprehensive study of the RPE in the new model they have created, and the results offer some intriguing new insights into mechanisms of retinal degeneration in CLN3 disease. This study should be of significant interest to the field and to investigators studying retinal disease more widely.

The authors have thoroughly characterized the iPS cells before differentiating these into RPE cells. Similar to what is seen in CLN3 patients, the authors show reduced autofluorescence in the CLN3 patient derived cells as compared to the control cells. Further, the authors functionally examined POS uptake and demonstrated decreased POS phagocytosis and that there is a defect in binding of POS in CLN3 deficient cells, which supports a novel hypothesis regarding the role of RPE in the retinal degeneration seen in CLN3 patients and would explain the reduced autofluorescence that is seen in CLN3 patient RPE. Finally, the authors suggest that CLN3 is present in RPE microvilli and have shown that overexpression of CLN3 rescues the phagocytosis defect.

We thank the reviewer for their appreciation of the iPSC-RPE data and for acknowledging that "this study should be of significant interest to the field and to investigators studying retinal disease more

widely.” We also thank the reviewer for the comments that have helped improve the data and presentation as detailed below.

Specific Comments

1. The CLN3 antibody data are not convincing and lack several key controls. As shown by Ezaki et al., 2003, J Neurochem, v87, 1296-1308, the CLN3 protein is highly glycosylated and this is reflected in its mobility on SDS-PAGE, showing up as a broad band between ~50 and 60 kDa, depending on the tissue source, due to differential glycosylation. Unfortunately, the authors have not shown any molecular weight standards on their blots throughout the manuscript, so the reader cannot see where the band shown as ‘CLN3’ is running. More importantly, the band shown is tight and not consistent with the above mentioned, published data. While the authors performed peptide preadsorption controls, the key controls of 1) blot data on lysates prepared from the Cln3 knockout mouse, showing the absence of the band believed to represent CLN3 and 2) blot data on the overexpressed protein, to show that indeed the antibody labels CLN3, are absent. The immunofluorescence data shown in Figure 7E show micrograph images from RPE from a wildtype and Cln3 ko mouse, but given the high level of autofluorescence in RPE, which is reduced when CLN3 is absent, these data too are not convincing without key controls such as no primary controls.

We agree with the reviewer that we should have shown the molecular weight on the Western blotting data. We apologize for this oversight and have now included relevant molecular weights on all the blots throughout the manuscript (Fig. 1, 4, 7, 8, S6, S8, S10). With regard to the CLN3 band, we observed a strong band ~50 kda when 15-20 µg total protein was loaded on gels for iPSC-RPE and native human RPE tissue (Fig. 7G, S10), although in some experiments when higher amount of total protein (30 µg) was loaded on gels, we did see an additional band consistent with CLN3 glycosylation (Fig. S10). In contrast, when protein isolated from native mouse RPE was run on SDS-PAGE, we observed multiple bands (due to non-specific background due to use of mouse CLN3 antibody) but the expected broad band between ~ 50-60 kda (1) was also seen (Fig. 7G). Furthermore, as the reviewer pointed out, the published literature, especially using mouse tissues and overexpression system utilizing mouse CLN3 has shown a broad band between ~ 50 and 60 kda (1). In fact, mouse and human CLN3 have been shown to run differently on Western blots and differential glycosylation has been seen based on species as well as the tissue source (1-3). Also, given that different glycosylation pattern is associated with human vs. mouse CLN3 (1, 2), and CLN3 has been shown to appear as a distinct band in human lymphoblast (2), we were not concerned with the results obtained by the CLN3 antibody in our Western analyses, especially given that when used on mouse RPE, the CLN3 antibody gave the expected results. However, we agree with the reviewer that this data needed to be in the manuscript and have now included it in Fig. 7 and S10 (also included in the appendix of this document).

With regard to the recommendations for antibody validation, in Fig. 7E we have now included the data on no primary control in the immunohistochemical analyses of mouse RPE sections. Of note, as detailed in the methods section because we used mouse-CLN3 primary antibody, we used mouse on mouse (M.O.M kit, Vector Lab Inc. FMK-2201) to overcome endogenous IgG staining for immunohistochemical analyses in parallel experiments that utilized no primary control, WT and CLN3 knockout mouse RPE. In retrospect we should have included the no primary control data in the original submission and are thankful to the reviewer for bringing this to our attention. With regard to antibody verification with Western blotting, as described earlier, the mouse Westerns yielded several bands (possibly due to use

of primary mouse antibody) and due to limited access to knockout tissue, we prioritized assessment of antibody by immunolocalization studies using MOM kit. However, as suggested by the reviewer, we have now included the data showing recognition of overexpressed CLN3 protein by the CLN3 antibody by Western blotting in Fig. S10. Of note, with regard to this data, we had to use MYC-CLN3 instead of FLAG-CLN3 as presence of FLAG-tag but not MYC-tag in the N-terminus interfered with CLN3 recognition by CLN3 antibody that is specific for an epitope mapping between amino acids 13-40 at the N-terminus of CLN3. Also, with regard to FLAG-CLN3, GFP expression by Western and FLAG/GFP expression by immunocytochemistry was used to confirm/assess transduction efficiency and these data are included in Fig. 7C, 7D, 8B, 8D, S8 & S9.

Also, we would like to point out due to the CLN3 antibody recognizing Myc-CLN3 but not FLAG-CLN3, we have used both MYC-CLN3 and FLAG-CLN3 to confirm the impact of CLN3 overexpression on rescue of POS phagocytosis defect in CLN3 disease hiPSC-RPE cells (Fig. 8). These changes have resulted in substantial text change in the results section. Please see results section under the subheading "A proportion of CLN3 is localized in the RPE microvilli in both native human and mouse tissue and hiPSC-RPE cells". Of note, the figure changes associated with these edits, Fig. 7, 8 and S10 are also presented in the appendix of this document.

2. Detail in the Figure legends are often lacking. For example, in Figure S7, are cells transduced or is endogenous CLN3 stained? If the latter, why is immunoreactivity not homogeneous throughout the cell layer? What antibody is used for the staining if transduced cells; is it FLAG or CLN3 antibody? Lamp1 staining has no characteristic punctuated pattern (most likely due to oversaturated?). Fluorescence in green channel overexposed. Lamp1 staining in CLN3 disease hiPSC-RPE cells in comparison would also be informative. The authors should clarify and should carefully review Figure legends throughout to ensure sufficient detail are provided to the reader to understand the experiments and the data. See additional comments below noting figure legends that require further detail.

We apologize for the lack of detail in the figure legends and have now edited all the figure legends throughout the manuscript to include appropriate details and increase clarity. We have also now included a more representative image showing LAMP1-CLN3 co-localization staining in FLAG-CLN3-GFP transduced hiPSC-RPE cells (Fig. S9). In addition, we have now included details for the antibody used in all figures showing data on transduced cells (Fig. 7, 8, S8 & S9). We also agree with the reviewer that Lamp1-CLN3 comparison in control vs. CLN3 disease iPSC-RPE cells would be informative and as mentioned in response to reviewer 1's comment #1, this will be the focus of a subsequent study where we evaluate the endosomal-lysosomal pathway in the CLN3 disease hiPSC-RPE model.

Of note, I am also including an image here showing the LAMP1 localization in control hiPSC-RPE cells to show that we do indeed see the expected punctate pattern (scale bar = 10 μ m).

3. Antibody details (catalog numbers) should be provided in the Methods.

We have now included catalog# for all antibodies in the Methods section.

4. RHO should be defined when it first appears in the manuscript.

We apologize for this oversight and have now defined RHO (Rhodopsin) and all other gene/protein symbol at the first mention.

5. Figure 8: Figure legend not comprehensive enough. Figure cannot be understood without having the main text at hand.

DAPI_LAMP1

As stated above for concern 2, we have now edited the figure legends to increase the level of detail/clarity. The revised Fig.8 and the corresponding figure legend are also presented in the appendix of this document. Of note, we have also revised the text pertinent to this data in the results subsection to "...lentivirus-mediated overexpression of wild-type CLN3 can rescue POS phagocytosis defect in CLN3 disease hiPSC-RPE cells" to clearly emphasize the results/data shown.

6. Fig 8A: Is endogenous CLN3 measured or transduced CLN3?

This is endogenous CLN3 compared in untransduced control vs. CLN3 disease hiPSC-RPE cells and we have now clarified this in both the results to state "Consistent with previously published studies (4, 5), quantitative real-time PCR analyses showed reduced expression of the endogenous CLN3 gene transcript in control vs. CLN3 disease hiPSC-RPE cultures (Fig. 8A) as well as in the Figure legend of Fig. 8 "Qualitative gel electrophoreses images (top panel) and quantitative real-time PCR analyses and (bottom panel) showing reduced expression of endogenous CLN3 gene in control vs. CLN3 disease hiPSC-RPE cells (n = 3). GAPDH served as loading control."

7. Fig 8 C, D: Scale bar would be beneficial to compare both images with each other. GFP-expression in transduced cells in B looks cytosolic, whereas it does look clustered in panel D. GFP expression is expected to be cytosolic.

We have now included the scale bar for both images. However, we do not always see GFP expression as cytosolic and this is because the bi-cistronic vector pHIV-Myc-CLN3-IRES-GFP, containing MYC tagged CLN3 and EGFP are translated separately by initiating the translation within the single mRNA at the IRES. The diffusion across and accumulation of enhanced GFP in the nucleus has been reported in various cell type (6). Studies have shown that homomultimeric forms of eGFP can diffuse into the nucleus due to its low molecular weight (~27 kDa) and due to the nuclear and kinetic entrapment of its homomultimers (7). We agree that this point needs to be clarified in the manuscript and have now added additional images in Fig. 7C, 8B, 8D, and S8 as well as the following text to the results section "Consistent with robust CLN3 overexpression, transduction of 293FT and control hiPSC-RPE cells with lentiviral vectors i) pHIV-MYC-CLN3-IRES-EGFP (Fig. S8) and ii) pHIV-FLAG-CLN3-IRES-EGFP (Fig. 7C, Fig. S8) resulted in prominent eGFP expression in both 293FT and hiPSC-RPE cells (Fig. 7C, 7D & Fig. S8). Of note, diffusion across and accumulation of eGFP in the nucleus after lentiviral

transduction (Fig. 7C & Fig. S8) is because the bi-cistronic vector pHIV-Myc-CLN3-IRES-GFP and pHIV-FLAG-CLN3-IRES-GFP containing CLN3 and EGFP are translated separately within the single mRNA at the internal ribosome entry site (IRES) (55). Of note, the presence of EGFP in the nucleus is due to its low molecular weight (~27 kDa) and the nuclear and kinetic entrapment of EGFP homomultimers (56)."

Fig 8 D, E: Which cells are patient RPE cells, which are from controls? (Do not understand the sentence 'the amount of RHO is still lower in CLN3 transduced hiPSC-RPE cells when compared to control'. Does it correlate with what is shown in the figure?).

*There was a typo in the figure legend that resulted in the confusion. These figures are only comparing POS uptake between untransduced and transduced CLN3 disease hiPSC-RPE cells and we have now fixed the text in the figure legend to address this issue and also included an additional panel Fig. 8G showing quantitative analyses of RHO and ACTN in untransduced vs. transduced cells. The revised figure legend states "E-G Representative Western blot images (E, F) and Western blot analyses post 2h POS feeding showing increased amount of RHO (monomer band 35 kDa, dimer band 70 kDa and aggregate/multimer bands >70 kDa) relative to total protein, (E, F), but similar levels of ACTN (G) relative to total protein in CLN3 disease hiPSC-RPE cells expressing WT-CLN3 (cells transduced with either pHIV-MYC-CLN3-IRES-EGFP and pHIV-FLAG-CLN3-IRES-EGFP lentiviral vectors) when compared to untransduced CLN3 disease hiPSC-RPE cells (n=4). Of note, as expected GFP band was observed only in transduced CLN3 disease hiPSC-RPE cells (F). Furthermore, as bicistronic lentiviral pHIV-MYC-CLN3-IRES-EGFP and pHIV-FLAG-CLN3-IRES-EGFP vectors were used in these experiments, the GFP band is seen at~ 25 kDa. *p< 0.05." The updated Fig. 8 is also presented in the appendix of the rebuttal letter.*

Fig 8 D, E: CLN3 or GFP WB band could be shown in addition to rhodopsin in the same blot/ for same samples to show equal transduction of CLN3 disease cells vs. controls.

As stated above for concern 9, there was confusion due to the typo in the text/legend and we have now included the GFP band to demarcate the untransduced vs. transduced CLN3-disease hiPSC-RPE cells.

8. Is POS isolated from WT? Was this experiment also tried with POS with CLN3-deficiency? This would add value to the study and strengthen the hypothesis regarding mechanism of retinal degeneration. Additional detail are also required in the Methods for this.

The POS is isolated from bovine eyes and is the usual protocol used to evaluate POS phagocytosis by primary and hiPSC-RPE cells (8). Of note, POS from hiPSC-derived cells are still not differentiated consistently and in quantities sufficient for POS phagocytosis assay (9) and thus were not utilized for comparing the impact of CLN3 dysfunction. Similarly, a large number of mouse eyes (>40-50) would be required to accomplish a single POS phagocytosis experiment and thus have been a limitation for comparing the impact of mutant POS on POS phagocytosis. Of note, we have now edited the methods section to further include details on POS used in this study to state that "Bovine POSs were obtained commercially from InVision BioResources (Cat. #98740, Seattle, WA). POSs from slaughterhouse retinas have previously been utilized by numerous studies to evaluate POS phagocytosis by human primary cells and hiPSC-RPE cells in culture (41, 43, 102-104)."

9. Abbreviation for EZR is not stated

We apologize for this oversight and have now defined EZR (EZRIN) and all other gene/protein symbol at the first mention.

10. Figure 1, Panel F: Twice labeled (Ff)

This has been fixed and thank you letting us know.

11. Primer used for RT-PCR experiments not listed (in particular those used for CLN3)

We have now included a supplementary table (Table S1) that lists all the primers for RT-PCR experiments, including CLN3. Table S1 is also presented in the appendix of this rebuttal letter.

12. Authors could explain more specifically how Microvilli length, width was analyzed. Manually or automated?

The microvilli were characterized in a blind fashion manually at UCLA and we have now included the details on microvilli quantification in the methods section to state, "For assessment of microvilli width, length and density, TEM images showing cells in longitudinal section were analyzed using Image J software (NIH) and Microsoft Excel. All measurements were done manually. Microvilli width was measured from outer membrane to outer membrane, about halfway up on 10 microvilli throughout a single image. On each image the length of the longest microvilli was evaluated, while the microvilli density was calculated by measuring the length of the RPE surface in each section and counting the number of microvilli. Using these measurements, the average number of microvilli per um of RPE length was determined. A total of about 100 images were analyzed this way, with 10 images from 5 different samples each for control and CLN3 disease hiPSC RPE cultures, respectively."

13. In the Discussion, the statement "...we utilized WT-CLN3 overexpression in CLN3 disease hiPSC-RPE cells to confirm a specific and direct role of CLN3 in POS phagocytosis" is an overinterpretation of what was shown. A direct role, particularly given the problems with the data on the CLN3 antibody, has not been shown. Therefore, it is suggested to revise this statement.

Although we have now included additional data on antibody verification, we agree with the reviewer and have revised the statement to read "we utilized WT-CLN3 overexpression in CLN3 disease hiPSC-RPE cells to investigate the role of CLN3 in POS phagocytosis."

14. In the Discussion, because the CLN3 antibody and localization data are unconvincing, the following statement should also be revised "In this study, we show that CLN3 is localized to apical microvilli of RPE and is essential for crucial structure (RPE microvilli) and function (POS phagocytosis) of RPE cells that are vital for photoreceptor survival and therefore vision."

Again, we have now included additional data on antibody verification for immunohistochemical analysis. However, in view of the reviewer's recommendation, we have revised the statement to read "we propose that CLN3 is localized to apical microvilli of RPE and is essential for crucial structure (RPE microvilli)

and function (POS phagocytosis) of RPE cells that are vital for photoreceptor survival and therefore vision.”

15. In the Discussion, it would be useful for the reader and would strengthen the overall manuscript if the authors included in the Discussion what is known about how RPE apical microvilli are formed/turned over and how this might relate to what is known about CLN3 function in the literature.

In accordance with the reviewer’s recommendation, we have now included the following sentences in the discussion “Of note, although there is currently limited information on the formation or turnover of RPE microvilli, the RPE microvilli are comprised of densely packed actin filaments and actin-associated proteins, including EZR and Ezrin-Radixin-Moesin-Binding Phosphoprotein 50 (EBP50) (75, 76). Notably, the amount of F-actin or EZR can impact microvillar formation (70, 77), and impact the phagocytic ability of the RPE by affecting POS binding ((78), (42)). Furthermore, cytoskeletal rearrangement is crucial for POS ingestion by RPE cells (79). Given that CLN3 has been shown to i) interact with non-muscle myosin IIB (NMIIB) (72), an actin-based motor protein that has been suggested to play a role in POS phagocytosis (80), and ii) impact cytoskeletal re-organization in other cell type(s) (80), it is plausible that CLN3 could impact microvillar formation and turnover as well as POS phagocytosis in RPE cells through its impact on F-actin and its associated proteins, including EZR and NMIIB, but possibly also others, such as RAC1 (81) and the Usher 1B protein, MYO7A (82, 83).”

Reviewer 3:

Overall comment:

Mutations in CLN3 can lead to loss of photoreceptors. Overall, this is an extensive piece of work with novel and interesting insights. This manuscript reports that CLN3 is required for phagocytosis of photoreceptor (PR) outer segments (POS) by retinal pigment epithelium (RPE) cells, which is essential for PR survival. Patient derived iPSC-RPE cells show defects in microvilli density and reduced POS binding and ingestion.

They report some CLN3 in RPE cells is localized to RPE microvilli, where phagocytosis occurs. Interesting, and relevant to therapeutic development, the phagocytic defects of RPE cells can be rescued by supplying CLN3 gene.

The paper ends with longitudinal imaging of the retina in a CLN3 disease patient which suggests that the autofluorescent changes and a POS phagocytosis defect may precede photoreceptor cell loss in CLN3 disease.

An interesting question from the therapeutic perspective is whether targeting RPE alone is sufficient to rescue loss of PRs, or if targeting PRs, do the RPE also have to be targeted?

Copious data is presented in a clear and logical manner.

We appreciate the reviewer’s enthusiasm for the study and are a grateful for the acknowledgement that the data was presented in clear and logical manner. We agree that the findings in this paper have therapeutic implications for photoreceptor cell loss in CLN3 disease and it is certainly a future follow up for cell/gene replacement therapy approach for CLN3 disease.

Specific Comments

1. CLN3-Batten is a new and therefore unusual terminology for this disease – I recommend using one that is more widely recognized.

Based on the reviewer's recommendation and to be current with the nomenclature, we have replaced "CLN3-Batten" with "CLN3 disease" throughout the manuscript.

2. Juvenile CLN3 disease is the most common type of NCL (as discussed later, other rare types of CLN3 disease may not lose vision, or not show other symptoms besides visual loss).

We agree with the reviewer that other NCLs might not have vision loss and CLN3 mutations can be associated with isolated retinal degeneration and have now further emphasized this point in the revised discussion to state: "Altogether, these results illustrate novel role of CLN3 in regulating POS phagocytosis in human RPE cells and i) suggest a role of primary RPE dysfunction in CLN3-associated retinal degeneration and ii) indicate gene-therapy targeting RPE cells as a potential treatment option to suppress photoreceptor cell loss in CLN3 disease caused due to the common 966 bp deletion."

What CLN3 antibody was used? How was it derived? What part of CLN3 does it recognize (N terminus, amino acids 13-40)? (See Fig S7). Part of M&M implies it is from Santa Cruz Technology, but this is not clear when all Abs are first listed together.

The antibody used is from Santa Cruz (SC-398192) and binds at the N-terminus, AA 13-40. In the Methods section, we have now included the catalog number for CLN3 antibody as well as all the other antibodies used in this study.

3. CLN3 is shown to colocalizes with lamp1 in hiPSC-RPE cells cells (Fig S7). Has the same experiment been done in these same cells? Ie does it also colocalise with lamp1? (Could some be mislocalised when over expressed?)

We agree with the reviewer that Lamp1 co-expression with CLN3 could be a consequence of CLN3 overexpression but the goal of LAMP1 co-localization in the current study was just to show that similar to previous overexpression studies, CLN3 overexpression leads to LAMP1-CLN3 co-localization in hiPSC-RPE cells and is concordant with CLN3 overexpression in mammalian cells (10). As stated in response/suggestion of reviewer 1 (comment # 1) and reviewer 2 (comment #2), we have included a paragraph in the discussion to highlight that endosomal/lysosomal function of CLN3 might be impacted in CLN3 disease and contribute to disease development and therefore evaluating this pathway, including co-localization of CLN3 and endosomal/lysosomal marker in human/hiPSC-RPE cells will be an important future direction in understanding CLN3 disease pathophysiology.

4. Fig 7 uses eyes from a mouse model that is no longer commonly used (Katz et al 1999), and certainly whose deletion does not match the patient derived cells used – why use this one? How old were the mouse from which the eyes were taken? At what point in the disease?

We agree with the reviewer that the CLN3 knockout mouse model (11) used here would not be appropriate for studying the disease pathophysiology and accordingly we only utilized it for confirming the antibody specificity and showing that unlike wild-type mouse RPE sections, staining for CLN3 in the RPE is not observed in the CLN3 knockout mouse at ~6 months of age (Fig. 7). We have now included

the age of the mice in methods section and emphasized this point in the results section further to state: “Of note, for immunocytochemical analyses, due to the debated specificity of previously used CLN3 antibodies (11), we utilized i) CLN3 blocking peptide (Fig. S10) ii) tissue from CLN3 knockout mice (58) (Fig. 7E) and iii) no primary antibody control (Fig. 7E). Of note, because we used a mouse-CLN3 antibody in our experiments, we also utilized a commercial kit (mouse on mouse or MOM kit, Vector Lab Inc. FMK-2201) to overcome the non-specific background due to endogenous IgG staining (59). Furthermore, the utility of retinal/RPE tissue from the CLN3 knockout mice (58) in the current study was limited to antibody verification for immunolocalization experiments.”

5. It is not clear which cells were used – M&M suggest there were 3 controls and 2 patient fibroblast lines used to derive iPSC. Presumably representative lines are shown in eg fig 1, 2, 3 and more. How are these chosen? Are the results consistent across all individual lines and clones, and controls? This is important given the genetic variation between lines beyond whether they have a mutation in CLN3.

We agree with the reviewer and have now clarified this point in the experimental set up section to state: “Unless stated otherwise, two distinct CLN3 disease iPSC lines both harboring the common 966 deletion in CLN3 gene and three distinct control hiPSC lines including familial heterozygote control and healthy unaffected control that did not harbor the mutation in CLN3 gene were used for all experiments. Additionally, throughout the study, experiments utilized two different clones from each of these hiPSC lines. Importantly, data presented in the manuscript arises from results that have been consistent across individual clones of each patient line and between the two patient lines. Furthermore, for all experiments with the exception of immunocytochemistry for RPE markers, parallel age-matched cultures of control and CLN3 disease hiPSC-RPE monolayer grown in Transwell inserts with TER > 150 Ω .cm⁻² (reported in vivo threshold, (47, 48)) were utilized. For immunocytochemical analyses of RPE markers, hiPSC-RPE on both transwells and coverslips were used.”

Important to emphasize that what is observed is specific to the consequences of the 966b deletion of CLN3. Another mutation of the CLN3 gene may give a different response.

We agree with the reviewer and as stated for comment # 2, we have now included a sentence to specify this in the discussion “Together, these results indicate a novel role for CLN3 in the apical RPE, and i) suggest that this apical function is primary in RPE dysfunction in CLN3-associated retinal degeneration and ii) indicate gene-therapy targeting RPE cells as a potential treatment option to suppress photoreceptor cell loss in CLN3 disease caused due to the common 966 bp deletion.”

6. *Western blots:*

In previous research papers, both cytoskeleton and glycolysis have been shown to be affected by the 1kb mutation and loss of CLN3. Thus there is concern about using ACTIN or GAPDH as a suitable normalisation marker. It is suggested that there was no difference in phalloidin staining but it looks more disperse compared to the control, plus it wasn't quantified. Is it possible to normalise to the whole protein level in the lysate to give a more accurate representation. Or show no changes in the same cell line when CLN3 is KD, removed by CRISPR or upregulated. Also bands look pixelated – why is this (compare ACTIN RHO WB in fig4B and fig4 D, fig 7B with EZR).

Based on the reviewer's recommendation, we have now included graphs showing quantitative analyses for specific protein levels (e.g. RHO, ACTN) normalized to whole protein level in the lysate throughout the manuscript (Fig. 4, 7, 8 and S6 & S10). We apologize that the bands look pixelated but this is an artifact of the Azure imaging system used for imaging and pixilation can be seen when the intensity of band is lower- and to avoid post-processing image alterations, we did not alter the image post-acquisition to make it look better. However, we have now replaced the Western blot images in Fig. 4B and 4E that were saved differently (tiff format) during acquisition. The updated Figures (Fig. 4, 8 S6, & S10) are also presented in the appendix of this rebuttal letter.

7. Significance – the 2nd sentence is not correct grammatically.

We appreciate the reviewer notifying us of this grammatical mistake and we have made the appropriate correction for the sentence to read “Mutations in CLN3 cause CLN3 disease, the most common form of NCL, a childhood onset disease characterized by premature death and progressive neurological and visual dysfunction.”

Once again, I would like to thank the reviewers for their valuable suggestion and critiques that have significantly helped improve the quality of the revised manuscript. If you have any questions, please contact me via email (ruchira_singh@urmc.rochester.edu) or phone (585-276-7338).

Sincerely,

Ruchira Singh, Ph.D.

References:

1. J. Ezaki *et al.*, Characterization of Cln3p, the gene product responsible for juvenile neuronal ceroid lipofuscinosis, as a lysosomal integral membrane glycoprotein. *J Neurochem* **87**, 1296-1308 (2003).
 2. D. A. Persaud-Sawin *et al.*, Neuronal ceroid lipofuscinosis: a common pathway? *Pediatr Res* **61**, 146-152 (2007).
 3. A. A. Golabek *et al.*, Expression studies of CLN3 protein (battenin) in fusion with the green fluorescent protein in mammalian cells in vitro. *Mol Genet Metab* **66**, 277-282 (1999).
 4. S. L. Cotman, J. F. Staropoli, The juvenile Batten disease protein, CLN3, and its role in regulating anterograde and retrograde post-Golgi trafficking. *Clin Lipidol* **7**, 79-91 (2012).
 5. S. L. Cotman *et al.*, Cln3(Deltaex7/8) knock-in mice with the common JNCL mutation exhibit progressive neurologic disease that begins before birth. *Hum Mol Genet* **11**, 2709-2721 (2002).
 6. S. K. Jang *et al.*, A segment of the 5' nontranslated region of encephalomyocarditis virus RNA directs internal entry of ribosomes during in vitro translation. *J Virol* **62**, 2636-2643 (1988).
 7. R. J. Jackson, M. T. Howell, A. Kaminski, The novel mechanism of initiation of picornavirus RNA translation. *Trends Biochem Sci* **15**, 477-483 (1990).
 8. C. Parinot, Q. Rieu, J. Chatagnon, S. C. Finnemann, E. F. Nandrot, Large-scale purification of porcine or bovine photoreceptor outer segments for phagocytosis assays on retinal pigment epithelial cells. *J Vis Exp* 10.3791/52100 (2014).
 9. K. J. Wahlin *et al.*, Photoreceptor Outer Segment-like Structures in Long-Term 3D Retinas from Human Pluripotent Stem Cells. *Sci Rep* **7**, 766 (2017).
 10. A. Kytala, G. Ihrke, J. Vesa, M. J. Schell, J. P. Luzio, Two motifs target Batten disease protein CLN3 to lysosomes in transfected nonneuronal and neuronal cells. *Mol Biol Cell* **15**, 1313-1323 (2004).
 11. M. L. Katz *et al.*, A mouse gene knockout model for juvenile ceroid-lipofuscinosis (Batten disease). *J Neurosci Res* **57**, 551-556 (1999).
-

Appendix

New Figures/ Figure panels

Fig. 2A

Fig. 2A) Autofluorescence overlying brightfield images in cryosections of mid-periphery fragments of retina-RPE-choroid obtained from a CLN3 disease donor and age-matched control eye showed decreased accumulation of autofluorescent material in the RPE layer of CLN3 disease donor eyes in spectral wavelength consistent with lipofuscin (red and green channels). Additionally, increased accumulation of autofluorescent debris is seen in the photoreceptor layer (indicated by white arrowheads) in the CLN3 disease donor retina compared to control retina. (Scale bar =20 μm) (n=1).

Fig. 9

Figure 9. Longitudinal multimodal imaging of a CLN3 disease patient is consistent with autofluorescent changes in photoreceptor-RPE complex preceding photoreceptor cell loss. Multimodal imaging of a normal eye (**A, B, C**) Fundus photograph of a normal right eye (**A**). Dashed circle outlines roughly the macula. The fovea is marked with an arrowhead in the center of the macula. Blood vessel and optic disc are also labeled (arrows). Fundus autofluorescence (FAF) image of a normal right eye (**B**). Note the optic disc and the blood vessels are dark as they do not exhibit autofluorescence. Optical coherence tomography (OCT) section of the retina through the fovea (yellow arrow in **B**) is shown. **C**) The photoreceptor nuclei are packed in the hypo-reflective band (arrowhead). The retinal pigment epithelium (RPE) is the linear hyper-reflective band marked by an arrow. Photoreceptor outer segments (POS) are present in the area between the photoreceptor nuclei and the RPE as alternating hyper- and hypo-reflective linear bands (marked with an asterisk). Multimodal imaging in a 7-year-old patient with CLN3 disease (**D-I**). **D**) Fundus photograph showing bull's eye maculopathy (BEM) outlined with arrowheads. **E**) FAF showing a faint hyperautofluorescent annulus at the margin of BEM (arrowheads). **F**) OCT section through the fovea reveals central loss of the photoreceptor nuclear layer as well as the POS layer. The layers are present towards the periphery on either side of the fovea but the layers are thinner than normal (arrowheads). **G**) At follow-up at age 8 years, BEM has enlarged (arrowheads) **H**) Hyperautofluorescent annulus has become more diffuse (arrowheads in **H**); and the loss of POS and photoreceptor nuclei can be seen across the entire section of the retina (**I**) with a decrease in overall thickness of the retina as well.

Fig. 7

Figure 7. A proportion of CLN3 protein localizes to RPE microvilli in both i) control hiPSC-RPE cells with and without CLN3 overexpression and ii) primary human and mouse RPE cells. A) Representative confocal microscopy image of the orthogonal view of control hiPSC-RPE monolayer post immunocytochemical analyses with antibodies against CLN3 and EZR (an RPE microvilli protein) showing apical presence of endogenous CLN3 and co-localization of endogenous CLN3 and EZR (scale bar =10 μ m) (n \geq 3). **B)** Representative confocal microscopy image post microvilli isolation with lectin-agarose beads and immunocytochemical analyses with CLN3 and EZR antibody showing co-localization of endogenous CLN3 and EZR in the control hiPSC-RPE microvilli-bound to lectin-agarose beads. (scale bar =10 μ m) (n = 3). **C)** Representative confocal microscopy images post immunostaining with a GFP antibody showing robust expression of EGFP in control CLN3 hiPSC-RPE cells transduced with pHIV-FLAG-IRES-eGFP lentiviral vector (scale bar =10 μ m) (n \geq 3). Of note, the observed EGFP localization in the nucleus is due to the nuclear and kinetic entrapment of EGFP homomultimers, and has been previously reported (55, 56). **D)** Representative confocal microscopy images post immunocytochemical analyses with FLAG and CLN3 antibodies showing co-localization of FLAG-CLN3 in control hiPSC-RPE cells transduced with pHIV-FLAG-IRES-eGFP lentiviral vector (scale bar =10 μ m) (n \geq 3). Of note, nuclei was stained with DAPI and is excluded in the top panel showing the D image to better visualize the CLN3-EZR co-localization. In the bottom panel showing the orthogonal view, DAPI is included to illustrate the apical localization of both FLAG-CLN3 and EZR. **E)** Confocal microscopy analyses of mouse retina sections after immunocytochemical analyses with CLN3 and EZR antibody showed co-localization of endogenous CLN3 with EZR (top panel). Notably, CLN3 antibody fails to detect CLN3 expression but EZR can be visualized in the RPE cells of CLN3^{-/-} mice (**E**, middle panel). Furthermore, no specific CLN3-staining was seen in WT mouse retina sections in the no primary controls that excluded incubation with primary antibody (**E**, bottom panel). Of note, because the host of CLN3 antibody is mouse, we utilized mouse on mouse (M.O.M $\text{\textcircled{R}}$) kit in these experiments scale bar =10 μ m) (n \geq 1). **F)** Confocal microscopy images post immunocytochemical analyses showed co-localization of endogenous CLN3 and EZR in primary human RPE wholemounts and orthogonal view (scale bar =10 μ m) (n \geq 1). **G)** Representative western blot images showing presence of endogenous CLN3 protein (50 kDa) using CLN3 specific antibody in WT mouse, primary human and hiPSC-RPE samples. (scale bar =10 μ m) (n \geq 1) **G)** Western blot image showing a distinct pattern of CLN3 in mouse vs. human RPE cells. Specifically, multiple bands for CLN3 were seen in native mouse RPE (~15-20 μ g total protein) compared to human RPE (primary, hiPSC) possibly due to non-specific background due to the mouse CLN3 antibody.

Fig. S10

Figure S10. Validation of CLN3 antibody in primary, hiPSC and mouse RPE samples.

A) Representative confocal microscopy images after immunostaining with CLN3 and ezrin (EZR) antibodies showing strong presence of endogenous CLN3 and co-localization of endogenous CLN3 and RPE microvilli protein, EZR, in wild-type (WT) mouse RPE (left panel). Notably, CLN3 blocking peptide abolishes detection of endogenous CLN3 but not EZR in mouse RPE sections (scale bar = 10µm). (n ≥ 3). **B)** Representative Western blot image showing two bands of endogenous CLN3 at ~ 50 kda (consistent with CLN3 glycosylation) in primary human RPE and control hiPSC-RPE samples (~20 µg total protein loaded on SDS-PAGE for each sample) analyzed with CLN3 antibody. **C)** Representative Western blot images showing a strong band ~50 kda in primary human RPE samples (~15 µg total protein loaded on SDS-PAGE) (left panel) analyzed with CLN3 antibody. Furthermore, neutralization of CLN3-antibody with CLN3 blocking peptide abolishes the ~50kda band in the same Western blot (right panel). **D, E).** Representative Western blot images (D) and quantitative analyses (E) showing ~6.5-fold increase in levels of CLN3 protein relative to the total protein in the hiPSC-RPE cell lysate in MYC-CLN3 transduced control hiPSC-RPE cells when compared to untransduced control hiPSC-RPE cells. **F)** Representative western blot images of control hiPSC-RPE cell vs. microvilli fraction, post RPE apical microvilli isolation using lectin-agarose beads (52, 100), showing the presence of endogenous CLN3 in both the cell pellet and microvilli fraction. In contrast, and as expected, the RPE microvilli protein, EZR, is predominantly present in the RPE microvilli fraction (n=3).

Fig. 8

Figure 8. Lentiviral-mediated overexpression of wild-type (WT)-CLN3 increases the amount of POS phagocytosed by CLN3 disease hiPSC-RPE cells. **A)** Qualitative gel electrophoreses images (top panel) and quantitative real-time PCR analyses and (bottom panel) showing reduced expression of endogenous CLN3 gene in control vs. CLN3 disease hiPSC-RPE cells ($n = 3$). GAPDH served as loading control. **B)** Representative confocal microscopy images post immunostaining with GFP specific antibody showing robust expression of GFP in transduced CLN3 disease hiPSC-RPE cells (pHIV-FLAG-CLN3-IRES-EGFP). Scale bar = 20 μm . **C)** Transepithelial resistance (TER) measurements showing no adverse effect of transduction on epithelial integrity in CLN3 hiPSC-RPE cells transduced with either pHIV-MYC-CLN3-IRES-EGFP and pHIV-MYC-CLN3-IRES-EGFP lentiviral vectors ($n=6$). **D)** Representative confocal microscopy images showing similar localization of tight junction protein, ZO-1, in transduced CLN3 disease hiPSC-RPE cells and untransduced CLN3 disease hiPSC-RPE cells. Of note, robust expression of GFP can be seen in CLN3 disease hiPSC-RPE transduced cells (pHIV-FLAG-CLN3-IRES-EGFP). Of note, GFP localization in the nucleus is due to the nuclear and kinetic entrapment of EGFP homomultimers (55, 56). Scale bar = 20 μm , ($n \geq 3$). **E-G)** Representative Western blot images (**E, F**) and Western blot analyses post 2h POS feeding showing increased amount of RHO (monomer band 35 kDa, dimer band 70 kDa and aggregate/multimer bands >70 kDa) relative to total protein, (**E, F**), but similar levels of ACTN (**G**) relative to total protein in CLN3 disease hiPSC-RPE cells expressing WT-CLN3 (cells transduced with either pHIV-MYC-CLN3-IRES-EGFP and pHIV-FLAG-CLN3-IRES-EGFP lentiviral vectors) when compared to untransduced CLN3 disease hiPSC-RPE cells ($n=4$). Of note, as expected GFP band was observed only in transduced CLN3 disease hiPSC-RPE cells (**F**). Furthermore as bicitronic lentiviral pHIV-MYC-CLN3-IRES-EGFP and pHIV-FLAG-CLN3-IRES-EGFP vectors were used in these experiments, the GFP band is seen at ~ 25 kDa. * $p < 0.05$.

Table S1:

1. List of primers pairs (5'-3') used for quantitative real-time PCR analyses.

Gene	Forward primer	Reverse primer
BEST1	ATTTATAGGCTGGCCCTCACGGAA	TGTTCTGCCGGAGTCATAAAGCCT
MERTK	AGCCTGAGAGCATGAATGTCACCA	TGTTGATCTGCACTCCCTTGGACA
MITF	TTCACGAGCGTCCTGTATGCAGAT	TTGCAAAGCAGGATCCATCAAGCC
PEDF	AGATCTCAGCTGCAAGATTGCCCA	ATGAATGAACTCGGAGGTGAGGCT
RPE65	GCCCTCCTGCACAAGTTTGACTTT	AGTTGGTCTCTGTGCAAGCGTAGT
OCCLUDIN	TCCTATAAATCCACGCCGGTTCCT	AGGTGTCTCAAAGTTACCACCGCT
CRALBP	TTCCGCATGGTACCTGAAGAGGAA	ACTGCAGCCGGAAATTCACATAGC
CLN3	ATCAGGGCGCGCATTGGAA	ACAGCAGCCGTAGAGACAGA
GAPDH	AGCAAGAGCACAAGAGGAAGAG	GAGCACAGGGTACTTTATTGATGG

2. Primer sequence used to detect the common 966 bp deletion in CLN3.

Forward: 5'- CATTCTGTCACCCTTAGAAGCC-3'

Reverse: 5'- GGCTATCAGAGTCCAGATTCCG-3'

Fig. 4

Figure 4. Reduced phagocytosis of unlabeled-POS by CLN3 disease hiPSC-RPE cells.

A) Schematic showing the protocol used to measure phagocytosis of unlabeled-POS by hiPSC-RPE cells. Specifically, hiPSC-RPE cultures were fed ~20-40 POS/RPE cells for 2h at 37°C. Subsequently, any remaining POS on the surface of RPE cells was removed by washing with 1X PBS and the amount of phagocytosed (bound + ingested POS) was quantified by measuring the amount of Rhodopsin (RHO), a POS-specific protein, within the hiPSC-RPE cells. Of note, the total protein in the hiPSC-RPE cell lysate served as the normalization control in these experiments. **B-E)**

Representative Western blot images (**B, E**) and quantitative Western blot analyses (**C, F**) post 2h POS-feeding showing reduced amount of RHO (monomer band 35 kDa, dimer band 70 kDa and aggregate/multimer bands >70 kDa that are normally seen in RHO Western analyses (35, 38) relative to total protein in parallel cultures of both young (D20-50 in culture) (**B, C**) and old (D60-150) (**E, F**) CLN3 disease hiPSC-RPE cells compared to control hiPSC-RPE cells. Of note, unlike RHO, no differences in Actin (ACTN) levels relative to total protein were seen between parallel cultures of both young (**B, D**) and old (**E, F**) control vs. CLN3 disease hiPSC-RPE cells in these experiments. * $p < 0.05$; ** $p < 0.005$.

Fig. S6

Figure S6. Similar protein levels of RPE binding and engulfment receptor in control vs. CLN3 disease hiPSC-RPE cells. A, B) Representative Western blot images (A) and quantitative Western blot analyses (B) showing similar protein levels of RPE binding receptor, ITGA5, ITGB5, and RPE engulfment receptor, MERTK, relative to total protein levels in control vs. CLN3 disease hiPSC-RPE. Similarly, no difference in levels of ACTN relative to total protein levels was seen between control and CLN3 disease hiPSC-RPE. (n=6).

REVIEWERS' COMMENTS:

Reviewer #1 (Remarks to the Author):

The authors have addressed my concerns. This is an excellent study.

Minor revision to Fig. 9: the yellow arrow is missing from panel "B". All yellow arrows should be converted to simple lines. An arrow implies you are pointing to something, whereas a line would indicate the location of the OCT scan.

Reviewer #2 (Remarks to the Author):

The authors have been very responsive to the reviews and have submitted a more solid manuscript with much improved level of detail. This study represents an important advance for the field that will be of much interest.

Associate Professor

November 1, 2020

Dear Reviewers:

We are glad to know that the reviewers were satisfied with the previous version of the manuscript and no further experimental data was required in the current submission. In accordance with the reviewers' comments and minor edits listed by Reviewer 1, we have made changes to Figure 9 in the revised version of the manuscript "**A human model of Batten disease shows role of CLN3 in phagocytosis at the photoreceptor-RPE interface.**" No other changes were requested in the revised submission.

Point by point response to Reviewer comments.

Reviewer #1 (Remarks to the Author):

The authors have addressed my concerns. This is an excellent study.

Minor revision to Fig. 9: the yellow arrow is missing from panel "B". All yellow arrows should be converted to simple lines. An arrow implies you are pointing to something, whereas a line would indicate the location of the OCT scan.

We are glad that the reviewer approves the study and have made changes to Figure 9 in accordance with the reviewer's recommendation.

Reviewer #2 (Remarks to the Author):

The authors have been very responsive to the reviews and have submitted a more solid manuscript with much improved level of detail. This study represents an important advance for the field that will be of much interest.

We are grateful to the reviewer for the acknowledging our efforts in addressing the reviewer's comments that have helped improve the study.

Once again, I would like to thank the reviewers for their valuable suggestion and critiques that have significantly helped improve the quality of the revised manuscript. If you have any questions, please contact me via email (ruchira_singh@urmc.rochester.edu) or phone (585-276-7338).

Sincerely,
